# ROBUST MODEL EVALUATION OVER LARGE-SCALE FEDERATED NETWORKS

## ABSTRACT

In this paper, we address the challenge of certifying the performance of a machine learning model on an unseen target network. We consider a source network "A" of $K$ clients, each with private data from unique and heterogeneous distributions, assumed to be independent samples from a broader meta-distribution $\mu$. Our goal is to provide certified guarantees for the model's performance on a different, unseen target network "B," governed by another meta-distribution $\mu'$, assuming the deviation between $\mu$ and $\mu'$ is bounded by either the *Wasserstein* distance or an *f-divergence*. We derive theoretical guarantees for the model's empirical average loss and provide uniform bounds on the risk CDF, where the latter correspond to novel and adversarially robust versions of the Glivenko-Cantelli theorem and the Dvoretzky-Kiefer-Wolfowitz (DKW) inequality. Our bounds are computable in polynomial time with a polynomial number of queries to the $K$ clients, preserving client privacy by querying only the model's (potentially adversarial) loss on private data. We also establish non-asymptotic generalization bounds that consistently converge to zero as both $K$ and the minimum client sample size grow. Extensive empirical evaluations validate the robustness and practicality of our bounds across real-world tasks.

## 1 INTRODUCTION

The distributed nature of modern learning environments, where training data and computational resources are scattered across clients in a network, has introduced challenges for the machine learning community. Federated learning (FL) addresses some of these challenges by enabling clients to collaboratively train a decentralized model through communication with a central server (McMahan et al., 2017; Liu et al., 2024). One major challenge is the heterogeneity of data distributions among clients. This non-IID nature of clients' data not only impacts the design of training algorithms but also complicates evaluation, particularly when models are applied to unseen clients from the same or different networks (Ye et al., 2023; Zawad et al., 2021).

In a standard FL setting, a fixed set of clients with a common learning objective trains a model that performs well on their data distributions. For example, in a mobile network, the goal is to generalize well to test data from observed clients. Most evaluations focus on average performance across the training clients. However, models are often applied to clients not involved in training (Reisizadeh et al., 2020). These unseen clients might have privacy concerns that prevent sharing any information, or they may be new clients who joined the network post-training (Ma et al., 2024). Furthermore, FL models trained on one network might later be tested on another network with different distributions. For example, a model trained on one city's clients may be applied to clients from another city. Therefore, it's crucial to evaluate models on unseen clients and networks beyond the original training clients.

In this work, we focus on evaluating FL models on unseen clients and networks. We assume access to clients whose data distributions are i.i.d. samples from a meta-distribution $\mu$, representing the client population (e.g., a mobile network in a specific region). Here, a *meta-distribution* models higher-level dynamics in the society that clients belong to, such as culture, lingual preferences, etc. Our goal is to provide performance guarantees for unseen clients whose data either follows $\mu$ or a different distribution within a bounded distance from $\mu$. The latter corresponds to a different society with slightly different styles, habits or preferences (see Figure 4 in Appendix E for a graphical

illustration). To address this challenge, we extend non-asymptotic average loss and CDF estimation guarantees to performance scores collected from observed clients. Specifically, we leverage the Dvoretzky–Kiefer–Wolfowitz (DKW) theorem to provide probably approximately correct (PAC) guarantees for unseen clients following $\mu$ (Dvoretzky et al., 1956). These guarantees offer lower and upper bounds on unseen clients' performance scores with adjustable confidence.

However, in real-world applications, test clients may come from different populations unseen during training. To account for this, we extend the DKW theorem to cover meta-distributions within a bounded divergence from the source. This extension includes an upper bound on the CDF for meta-distributions within a bounded $f$-divergence. We also give performance guarantees via tightly bounding the meta-distributionally shifted average loss when both $f$-divergence or Wasserstein shifts are considered. The $f$-divergence approach captures potential reweightings of client types in the target network while Wasserstein distances model more complex distributional shifts (e.g., new clients from previously unseen regions) (Wang et al., 2022; Kuhn et al., 2019). To our knowledge, this is the first extension of the DKW and subsequently Glivenko-Cantelli theorems to such a robust setting.

Our bounds are asymptotically minimax optimal, meaning they cannot be improved, at least asymptotically. They are also consistent, converging to zero as the number of observed clients and data points per client increases. We further show that handling Wasserstein-type shifts can be achieved by querying clients for adversarial loss instead of ordinary loss, without requiring access to user data or centralizing data at the server. Additionally, we demonstrate that the number of client queries can be bounded by a polynomial function of the problem parameters. Lastly, we validate the practicality and computational efficiency of our bounds through experiments on real-world datasets.

## 2  RELATED WORK

**Evaluation and Generalization of models learned by FL algorithms on unseen clients.** The challenge of generalizing FL models to unseen clients and distributions has been studied in several related works. Li et al. (2020) introduce a method to address the heterogeneity of client data, focusing on improving the generalization of FL models. Similarly, Ma et al. (2024) propose a topology-aware federated learning approach that leverages client relationships to enhance model robustness against out-of-federation data. Also, Zeng et al. (2023) explore adaptive federated learning techniques to dynamically adjust model parameters based on client data distributions.

**Distribution shifts and adversarial robustness in FL.** The robustness of FL models against distribution shifts and adversarial attacks has been the focus of several related references. Reisizadeh et al. (2020) propose a robust federated learning framework to handle affine distribution shifts across clients' data. Their proposed framework incorporates a Wasserstein-distance-based distribution shift model to account for device-dependent data perturbations. Also, Zhang et al. (2023) conduct comprehensive evaluations on the adversarial robustness of FL models, proposing the decision boundary-based FL Training algorithm to enhance the the trained model's robustness. Zhou et al. (2022) gain insight from the bias-variance decomposition to improve adversarial robustness in FL. Also, Ben Mansour et al. (2022) propose a robust aggregation method to reduce the effect of adversarial clients.

**Performance guarantees in Heterogeneous FL.** The heterogeneous nature of FL, where clients have different data distributions has been a topic of great interest in the literature. Fallah et al. (2020) introduce a personalized federated learning framework based on model-agnostic meta-learning, which provides performance guarantees by optimizing for data distribution heterogeneity. Luo et al. (2021) propose a classifier calibration method that adjusts for bias in heterogeneous data, offering improved performance guarantees in non-IID settings. Tan et al. (2022) develop FedProto, a framework that leverages prototype learning to improve convergence and robustness under non-convex objectives.. Moreover, Wu et al. (2024) propose FedLoRA, which adapts low-rank parameter sharing techniques to mitigate the effects of heterogeneity in personalized federated learning. Also, Cheng et al. (2024) and Jia et al. (2024) introduce group-based customization and local parameter sharing strategies, respectively, to provide fairness and efficiency guarantees for heterogeneous client types and multiple tasks in FL.

## 3 NOTATIONS AND PRELIMINARIES

For $K \in \mathbb{N}$, $[K]$ denotes the set $\{1, 2, \ldots, K\}$. Consider two measurable spaces $\mathcal{X}$ and $\mathcal{Y}$, referred to as the feature and label spaces, respectively, where we typically have $\mathcal{X} \subseteq \mathbb{R}^d$ and $\mathcal{Y}$ typically $\{1, \ldots, c\}$. The joint feature-label space is $\mathcal{Z} \triangleq \mathcal{X} \times \mathcal{Y}$, and $\mathcal{M}(\mathcal{Z})$ denotes the set of all probability measures on $\mathcal{Z}$. We usually abbreviate $\mathcal{M}(\mathcal{Z})$ with $\mathcal{M}$, for simplicity. Therefore, each distribution $P \in \mathcal{M}(\mathcal{Z})$ represents a joint measure over a feature vector $\boldsymbol{X} \in \mathcal{X}$ and its label $y \in \mathcal{Y}$.

A meta-distribution $\mu \sim \mathcal{M}(\mathcal{M}(\mathcal{Z}))$ is a distribution over distributions, where samples from $\mu$ represent different domains (or distributions) over $\mathcal{X} \times \mathcal{Y}$. We usually abbreviate $\mathcal{M}(\mathcal{M}(\mathcal{Z}))$ with $\mathcal{M}^2$ for simplicity. For any measures $P, Q \in \mathcal{M}$, let $\mathcal{D}(P, Q)$ denote a distance or divergence between them. In Section 5, $\mathcal{D}$ is an $f$-divergence for a properly defined function $f(\cdot)$, and in Section 6, it is a Wasserstein metric. The $\varepsilon$-distributional ambiguity ball $\mathcal{B}_\varepsilon(P)$ is the set of distributions within $\varepsilon$ distance from $P$ according to $\mathcal{D}$:

$$\mathcal{B}_\varepsilon(P) \triangleq \{Q \in \mathcal{M} \mid \mathcal{D}(P, Q) \leq \varepsilon\}. \tag{1}$$

Let $h : \mathcal{X} \to \mathcal{Y}$ represent a given hypothesis (e.g., a classifier) and assume a fixed loss function $\ell : \mathcal{Y} \times \mathcal{Y} \to \mathbb{R}_{\geq 0}$. The expected *Risk* of $h$ with respect to $P \in \mathcal{M}$ is defined as $R(h, P) \triangleq \mathbb{E}_P \{\ell(y, h(\boldsymbol{X}))\}$. The adversarial risk for $h$ w.r.t. a measure $P$ and a radius $\varepsilon$ is formulated as $\sup_{Q \in \mathcal{B}_\varepsilon(P)} \mathbb{E}_Q [\ell(y, h(\boldsymbol{X}))]$, which represents the worst expected loss within the $\varepsilon$-neighborhood of $P$ according to $\mathcal{D}$.

Meta-distributional ambiguity balls $\mathcal{G}_\varepsilon(\mu)$ are the set of meta-distributions within an $\varepsilon$ distance from a base meta-distribution $\mu$, where the distance $\tilde{\mathcal{D}}$ can be an $f$-divergence or a Wasserstein metric. The transportation cost of the Wasserstein metric in the meta-distributional space, can itself be an ordinary Wasserstein metric between distributions in $\mathcal{M}$. The meta-distributionally robust loss of a classifier $h$ with respect to $\mu$ is defined as:

$$\sup_{\mu' \in \mathcal{G}_\varepsilon(\mu)} \mathbb{E}_{P \sim \mu'} [\mathbb{E}_P [\ell(y, h(\boldsymbol{X}))]].$$

In Section 5, we focus on $f$-divergence balls, while Wasserstein balls are covered in Section 6.

## 4 PROBLEM DEFINITION

This section describes our data generation process and specifies the privacy-preserving query policy that governs communication between clients and the server. After that, we formally define our problem.

**Data Generation and Privacy Constraints:** Consider $K \in \mathbb{N}$ clients connected to a central server. Each client $k \in [K]$ is associated with a unique and private data distribution model $P_k \in \mathcal{M}$, referred to as the $k$th *domain*. Let $P_1, P_2, \ldots, P_K$ be i.i.d. realizations of an unknown meta-distribution $\mu$ over $\mathcal{Z}$. However, client $k$ does not have direct knowledge of their corresponding domain $P_k$. Instead, they have access to a dataset $D_k$ of size $n_k \in \mathbb{N}$, which contains i.i.d. samples from $P_k$, i.e.,

$$D_k \triangleq \left\{ \left(\boldsymbol{X}_i^{(k)}, y_i^{(k)}\right) \right\}_{i=1}^{n_k} \overset{i.i.d.}{\sim} P_k^{\otimes n_k}. \tag{2}$$

Let $\widehat{P}_k$ denote the empirical version of $P_k$ based on the samples in $D_k$. It should be noted that i) server knows the value of $n_k$, ii) no one knows about $P_k$, and iii) $\widehat{P}_k$ is known, but only to client $k$.

**Server-Client Query Policy:** This procedure serves as the foundation for communication between the server and clients throughout the paper. The server can query each of the $K$ clients as follows: it sends a model $h$ and a robustness radius $\rho \geq 0$ to client $k \in [K]$. In response, the client returns the "Query Value", the adversarial risk of $h$ around $\widehat{P}_k$:

$$\widehat{\mathsf{QV}}_k(h, \rho) \triangleq \sup_{Q \in \mathcal{B}_\rho(\widehat{P}_k)} \mathbb{E}_Q[\ell(y, h(\boldsymbol{X}))]. \tag{3}$$

The type of distributional ball $\mathcal{B}_\rho(\cdot)$ can be defined using any application-specific divergence or metric over $\mathcal{M}$. In this work, we only use the adversarial regime (i.e., $\rho > 0$) in Section 6 which focuses on Wasserstein meta-distributional shifts. When the robustness radius is omitted, it means it is zero, i.e., $\widetilde{\mathsf{QV}}_k(h) = \widehat{\mathsf{QV}}_k(h, 0)$. Each client $k \in [K]$ has a maximum number of queries it will accept from the server, referred to as the *query budget*.

### 4.1 Our Problem Setup

In a large network with $K$ clients and a central server, assume an unknown meta-distribution $\mu \in \mathcal{M}^2$. The server sends a model $h$ to clients and requests various robust or non-robust risk values for different robustness radii. The goal is to compute a meta-distributionally robust upper bound for the average or CDF of the risk of $h$. Specifically, in the case of bounding the average risk the objective is to compute an empirical value $\widehat{E}(\varepsilon)$ such that with high probability we have:

$$\sup_{\mu' \in \mathcal{G}_\varepsilon(\mu)} \mathbb{E}_{P \sim \mu'} \left[ \mathbb{E}_P \left[ \ell(y, h(\boldsymbol{X})) \right] \right] \leq \widehat{E}(\varepsilon) + \zeta,$$

where $\widehat{E}(\varepsilon)$ can be efficiently computed by the server. The generalization gap $\zeta$ is expected to vanish as both $K$ and $\min_k n_k$ increase asymptotically. The shape of the meta-distributional ambiguity ball $\mathcal{G}_\varepsilon(\mu)$ can be tailored to the application; here, we consider $f$-divergence and Wasserstein balls. Algorithmically, $\widehat{E}(\varepsilon)$ is determined using the server-client query policy. The server controls the number of queries and the value of $\rho$ for each, ensuring that computational costs grow at most polynomially with problem parameters.

## 5 $f$-DIVERGENCE META-DISTRIBUTIONAL SHIFTS

In this section, we provide high-probability guarantees for the meta-distributionally robust performance of $h$, focusing on both the average loss and the CDF of the risk, under $f$-divergence-based meta-distributional shifts. $f$-divergence, for different choices of the convex function $f(\cdot)$ is a powerful tool for modeling variations between two probability measures that share the same support.

Definition of $f$-divergence can be found in Definition A.1 (see Appendix A). We also assume another useful property between source and target meta-distributions denoted by Density Ratio Boundedness property (see Assumption A.2 in Appendix A). This boundedness for the density ratios of the meta-distributions $\mu$ and $\mu'$ is required for a number of theoretical guarantees and practical algorithms, as it limits how much $\mu'$ can deviate from $\mu$, preventing extreme values that could destabilize learning models or theoretical analyses. At this point, we can define an $f$-Divergence ball in $\mathcal{M}^2$ as follows:

**Definition 5.1** (Meta-Distributional $f$-Divergence Ball). For a meta-distribution $\mu \in \mathcal{M}^2$, a function $f$ as defined in Definition A.1, $\varepsilon \geq 0$ and $\Lambda \geq 1$, the $f$-divergence ball $\mathcal{G}_{\varepsilon,\lambda}^{f-\mathrm{div}}(\mu)$ is defined as:

$$\mathcal{G}_{\varepsilon,\Lambda}^{f-\mathrm{div}}(\mu) \triangleq \left\{ \mu' \mid \mathcal{D}_f(\mu' \| \mu) \leq \varepsilon, \ \mu' \text{ and } \mu \text{ have } \Lambda \text{ bounded density ratio} \right\}. \tag{4}$$

It essentially describes a neighborhood around $\mu$ where the divergence does not exceed $\varepsilon$, providing a controlled way to manage deviations in distributional shifts.

### 5.1 Theoretical Guarantees on Robustness of Empirical Average Risk

The preceding definitions and assumptions collectively provide a structured way to handle distributional shifts using $f$-divergence. At this point, we can state our main result in the form of the following theorem:

**Theorem 5.2** (Bounding Empirical Risk with $f$-Divergence Robustness). *Assume a fixed and unknown meta-distributions $\mu \in \mathcal{M}^2$, let $\varepsilon, \delta > 0$ and $\Lambda \geq 1$, and consider $f$ to be as in Definition A.1. Let $h : \mathcal{X} \to \mathcal{Y}$ be an arbitrary model and $\ell : \mathcal{Y}^2 \to \mathbb{R}$ be any 1-bounded loss function. Assume $P_1, \ldots, P_K$ represent $K$ i.i.d. and unknown sample distributions from $\mu$. Accordingly, let $\{(\boldsymbol{X}_i^{(k)}, y_i^{(k)})\}_{i=1}^{n_k}$ for $k \in [K]$ represent a known (but private) empirical dataset of size $n_k \geq 1$ containing i.i.d. samples from $P_k$. Define $\widehat{B}^*$ as:*

$$\widehat{B}^*(\varepsilon) \triangleq \sup_{\Lambda^{-1} \leq \alpha_1, \ldots, \alpha_K \leq \Lambda} \frac{1}{K} \sum_{k \in [K]} \alpha_k \widehat{\mathsf{QV}}_k(h) \tag{5}$$

$$\text{subject to} \quad \left| \frac{1}{K} \sum_{k \in [K]} \alpha_k - 1 \right| \leq c_1 \sqrt{\frac{\log\left(\frac{1}{\delta}\right)}{K}} \ , \ \frac{1}{K} \sum_{k \in [K]} f(\alpha_k) \leq \varepsilon + \frac{c_2}{\sqrt{K}},$$

*where constants $c_1, c_2 \geq 0$ only depend on $\Lambda$ and $f(\cdot)$. Then, for any $\delta > 0$, the following bound holds with probability at least $1 - \delta$:*

$$\sup_{\mu' \in \mathcal{G}_{\varepsilon, \Lambda}^{f-\mathrm{div}}(\mu)} \mathbb{E}_{P \sim \mu'} \left[ \mathbb{E}_P \left[ \ell \left( y, h \left( \boldsymbol{X} \right) \right) \right] \right] \; \leq \; \widehat{B}^*(\varepsilon) + \sqrt{\frac{\log \left( \frac{K+3}{\delta} \right)}{2}} \left[ \Lambda K^{-1/2} + \frac{1}{K} \sum_{k \in [K]} n_k^{-1/2} \right].$$

Proof is given in Appendix B. This theorem provides a robust bound on the expected loss under meta-distributional shifts using $\widehat{B}^*(\varepsilon)$, which determines an upper bound on the average adversarial loss. With high probability (at least $1 - \delta$), the expected loss under any shifted meta-distribution $\mu'$ can be bounded by $\widehat{B}^*(\varepsilon)$ plus a term that decreases with increasing $K$ and $\min_{k \in [K]} n_k$, reflecting the amount of data/clients. The main advantage of our result is that our generalization gap does not have a *non-vanishing* $\varepsilon$-dependent term, unlike many existing bounds. Mathematically speaking, we proved the following for any $\mu, \mu'$ which are $\varepsilon$-proximal in the $f$-divergence sense:

$$\mathbb{E}_{\mu'} \left[ \mathbb{E}_P \left( \ell \left( y, h \left( \boldsymbol{X} \right) \right) \right) \right] - \widehat{B}^* (\varepsilon) \; \overset{w.h.p.}{\leq} \; \tilde{\mathcal{O}} \left( K^{-1/2} + \left[ \min_{k \in [K]} n_k \right]^{-1/2} \right), \qquad (6)$$

where $\tilde{\mathcal{O}}$ hides logarithmic dependencies. Additionally, any inherent "wellness" or "robustness" of the target model $h$ is directly reflected through a low value for $\widehat{B}^*(\varepsilon)$. In other words, if $h$ results in a *low* error/loss across the sampled distributions $P_1, \ldots, P_K \sim \mu$ (or their empirical counterparts), $\widehat{B}^*(\varepsilon)$ remains small as well, regardless of the value of $\varepsilon$. To the best of our knowledge, conventional bounds do not exhibit this adaptivity. Another notable fact is that $\widehat{B}^*(\varepsilon)$ is asymptotically minimax optimal, since when $K, \min_k n_k \to \infty$ it almost surely becomes the true (statistical) adversarial loss of $h$ due to the strong law of large numbers.

*Remark* 5.3 (**Computational Complexity**). The server-side program in equation 5 to determine the value of $\widehat{B}^* (\varepsilon)$ is convex. Given certain smoothness properties of the function $f(\cdot)$, a standard convex optimization algorithm approximates $\widehat{B}^* (\varepsilon)$ within an arbitrary error margin $\Delta > 0$, with polynomial time complexity relative to $K$ and $\Delta^{-1}$. Also, the total query budget is only one ordinary (non-robust) query per client.

## 5.2 Uniform Robustness of the Empirical Risk Distribution

For the case of $f$-divergence robustness, we show that it is also possible to derive an asymptotically consistent and *uniform* bound for the risk distribution (CDF) which estimates $\mu' \left( \mathbb{E}_P \left[ \ell \left( y, h \left( \boldsymbol{X} \right) \right) \right] \geq \lambda \right)$, for all $\lambda \in \mathbb{R}$, simultaneously. Here, $\mu'$ represents the unknown target meta-distribution, which is assumed to be within an $\varepsilon$-proximity of the source meta-distribution $\mu$. Recall that $\mu$ is also unknown, and our access to it is through $K$ i.i.d. empirical and private realizations $\widehat{P}_1, \ldots, \widehat{P}_K$ with sizes $n_1$ through $n_k$. To achieve this, we first derive a robust version of the uniform convergence bound on empirical CDFs, known as the Dvoretzky-Kiefer-Wolfowitz (DKW) inequality:

**Lemma 5.4** (Robust Version of DKW Inequality). *Let $\mu$ and $\mu'$ be two probability measures on $\mathbb{R}$ where $\mu'$ is absolutely continuous w.r.t. $\mu$. Assume $X_1, \ldots, X_n$ for $n \in \mathbb{N}$ to be i.i.d. samples drawn from $\mu$. Suppose we have $\mathcal{D}_f \left( \mu' \| \mu \right) \leq \varepsilon$ for some $\varepsilon \geq 0$ and a proper convex function $f(\cdot)$. For $\delta > 0$, let us define the set $\mathcal{A}_n \left( \varepsilon, \delta \right)$ as*

$$\mathcal{A}_n \left( \varepsilon, \delta \right) \triangleq \left\{ \boldsymbol{\alpha} \in \mathbb{R}_+^n \; \middle| \; \left| \frac{1}{n} \sum_{i=1}^n \alpha_i - 1 \right| \leq c_1 \sqrt{\frac{\log \left( \frac{1}{\delta} \right)}{n}}, \; \frac{1}{n} \sum_{i=1}^n f \left( \alpha_i \right) \leq \varepsilon + c_2 \sqrt{\frac{\log \left( \frac{1}{\delta} \right)}{n}} \right\},$$

*where constants $c_1$ and $c_2$ depend only on $f(\cdot)$. Then, there exists $\boldsymbol{\alpha} \in \mathcal{A}_n \left( \varepsilon, \delta \right)$ such that the following uniform bound holds with probability at least $1 - \delta$:*

$$\sup_{\lambda \in \mathbb{R}} \left| \mu' \left( X \leq \lambda \right) - \frac{1}{n} \sum_{i=1}^n \alpha_i \mathbb{1} \left( X_i \leq \lambda \right) \right| \leq \mathcal{O} \left( \sqrt{\frac{\log \left( \frac{n}{\delta} \right)}{n}} \right). \qquad (7)$$

Proof is given in Appendix B. A direct corollary of Lemma 5.4 is the robust (again with respect to $f$-divergence adversaries) of the well-known Glivenko-Cantelli theorem, which can be found in Appendix A. Building on the results from Lemma 5.4 and following a similar approach as in Theorem 5.2, we propose the following theorem which introduces our uniformly convergent and consistent estimator for the risk CDF:

**Theorem 5.5** (Empirical Risk Distribution with $f$-Divergence Robustness). *Assume the setting described in Theorem 5.2. For any $\lambda \in \mathbb{R}$ let us define:*

$$\widehat{J}^*(\varepsilon, \lambda) \triangleq \sup_{\Lambda^{-1} \leq \alpha_1, \ldots, \alpha_K \leq \Lambda} \frac{1}{K} \sum_{k \in [K]} \alpha_k \mathbb{1}\left(\widehat{\mathsf{QV}}_k(h) \geq \lambda - \sqrt{\frac{\log\left(\frac{K+2}{\delta}\right)}{2n_k}}\right) \tag{8}$$

$$\text{subject to} \quad \left|\frac{1}{K} \sum_{k \in [K]} \alpha_k - 1\right| \leq c_1 \sqrt{\frac{\log\left(\frac{K+2}{\delta}\right)}{K}} \ , \ \frac{1}{K} \sum_{k \in [K]} f(\alpha_k) \leq \varepsilon + c_2 \sqrt{\frac{\log\left(\frac{K+2}{\delta}\right)}{K}},$$

*where $c_1, c_2$ are constants that only depend on $f(\cdot)$ and $\Lambda$. Then, with probability at least $1 - \delta$, the following bound holds uniformly over all $\lambda \in \mathbb{R}$:*

$$\sup_{\mu' \in \mathcal{G}_{\varepsilon,\Lambda}^{f-\mathrm{div}}(\mu)} \mu'\left(\mathbb{E}_P\left[\ell(y, h(\boldsymbol{X}))\right] \geq \lambda\right) \ \leq \ \widehat{J}^*(\varepsilon, \lambda) + \mathcal{O}\left(\Lambda \sqrt{\frac{\log\left(\frac{K+2}{\delta}\right)}{K}}\right). \tag{9}$$

The proof is given in Appendix B. The bound in equation 9 exhibits the following properties: i) It is forward-shifted with respect to the true CDF, meaning it exhibits a delayed reaction to increasing $\lambda$ compared to the true CDF. The maximum delay is on the order of $\mathcal{O}(\sqrt{\log K / \min_k n_k})$. ii) The value of the CDF estimator also deviates from the true estimator by the amount $\mathcal{O}(\Lambda\sqrt{\log K/K})$. Consequently, as $K$ and $(\min_k n_k)/\log K$ tend to infinity, the bound becomes asymptotically tight. iii) The bound holds uniformly over all $\lambda \in \mathbb{R}$, similar to the original DKW inequality and its corresponding corollary, the Glivenko-Cantelli theorem.

It should be noted that although a separate maximization problem needs to be solved for each value of $\lambda$, these problems are all convex and can therefore be solved efficiently. Additionally, according to Lemma 5.4, there exists a single $\boldsymbol{\alpha} \in \mathcal{A}_K(\varepsilon, \delta)$ such that the bound holds for all $\lambda$. However, since this vector $\boldsymbol{\alpha}$ is not known, we must take the supremum over $\mathcal{A}_K$ for each $\lambda$, separately.

# 6 Wasserstein Meta-Distributional Shifts

This section addresses the case of Wasserstein meta-distributional shifts from $\mu$ to $\mu'$. Such shifts present significant challenges to our robust model evaluation framework, as the model $h$ may encounter entirely unseen regions of the distributional support in the target domain. In practical terms, this implies that users may exhibit data distributions that are entirely novel or even drastically different from those encountered during the training phase on the source network. We only provide theoretical guarantees for the average risk, but not for the entire risk distribution. Addressing the full risk distribution requires a more advanced methodology, which we defer to future work.

Let us begin by introducing new definitions related to both standard and meta-distributional Wasserstein metrics, and then present our main result, an analog of Theorem 5.2 for Wasserstein-type shifts. Wasserstein metric w.r.t. a proper transportation cost $c : \mathcal{Z}^2 \to \mathbb{R}$ is defined in Definition 6.1 (see Appendix A). In a similar fashion, one can define the Wasserstein distance between any two meta-distributions $\mu, \mu' \in \mathcal{M}^2(\mathcal{Z})$ with respect to any valid transportation cost over the space of measures $\mathcal{M}(\mathcal{Z})$, such as the ordinary Wasserstein distance. Mathematically speaking:

**Definition 6.1** (Wasserstein metric over $\mathcal{M}^2$). For any two meta-distributions $\mu, \mu' \in \mathcal{M}^2(\mathcal{Z})$, and a lower semi-continuous transportation cost $c : \mathcal{Z} \times \mathcal{Z} \to \mathbb{R}_{\geq 0}$, let us define

$$\|\mu - \mu'\|_{\mathcal{W}_c} \triangleq \inf_{\nu \in \mathcal{C}(\mu, \mu')} \mathbb{E}_{(P,Q) \sim \nu}\left\{\mathcal{W}_c(P, Q)\right\} \tag{10}$$

as the Wasserstein distance between $\mu$ and $\mu'$ w.r.t. transportation cost $\mathcal{W}_c$. Here, $\mathcal{C}(\mu, \mu')$ is the set of all couplings between $\mu$ and $\mu'$.

Accordingly, for $\varepsilon \geq 0$ and $P \in \mathcal{M}$, we define the Wasserstein ball of radius $\varepsilon$ around $P \in \mathcal{M}$ as $\mathcal{B}_{\varepsilon}^{\mathrm{wass}}(P) \triangleq \{Q|\, \mathcal{W}_c(P,Q) \leq \varepsilon\}$, where the transportation cost $c$ is hidden from formulation for the sake of simplicity. Similarly, one can define the Wasserstein ball around meta-distribution $\mu \in \mathcal{M}^2(\mathcal{Z})$ with radius $\varepsilon$ as the set of meta-distributions with a (meta-distributional) Wasserstein distance of at most $\varepsilon$ from $\mu$ as follows:

$$\mathcal{G}_{\varepsilon}(\mu) \triangleq \left\{ \mu'|\ \|\mu - \mu'\|_{\mathcal{W}_c} \leq \varepsilon \right\}. \tag{11}$$

We now present our main result of this section: a quasi-convex (and thus polynomial-time) optimization problem that provides empirical meta-distributionally robust evaluation guarantees against Wasserstein shifts, with a asymptotically vanishing generalization gap.

**Theorem 6.2** (Empirical Risk of $h$ with Wasserstein Robustness)**.** *Assume an unknown meta-distributions $\mu \in \mathcal{M}^2$, let $h : \mathcal{X} \to \mathcal{Y}$ be an arbitrarily hypothesis and let $\ell$ be a $1$-bounded loss function. Assume $P_1, \ldots, P_K$ represent $K$ i.i.d. and unknown sample distributions from $\mu$. Accordingly, let $\{(\boldsymbol{X}_i^{(k)}, y_i^{(k)})\}_{i=1}^{n_k}$ for $k \in [K]$ represent a known (but private) empirical dataset of size $n_k$ consisting of i.i.d. samples from $P_k$. Let $c$ be a bounded and proper (according to Definition 6.1) transportation cost on $\mathcal{Z} \times \mathcal{Z}$. For any given $\varepsilon, \delta > 0$, consider the following program:*

$$\widehat{U}^*(\varepsilon) \triangleq \sup_{\rho_1, \ldots, \rho_K \geq \varepsilon/K} \frac{1}{K} \sum_{k \in [K]} \widehat{\mathsf{QV}}_k(h, \rho_k)$$

$$\text{subject to} \quad \frac{1}{K} \sum_{k \in [K]} \rho_k \leq \varepsilon \left(1 + \frac{1}{K}\right) + c_1 \sqrt{\frac{\log\left(\frac{K+2}{\delta}\right)}{K}}. \tag{12}$$

*Then, the following bound holds with probability at least $1 - \delta$ for the meta-distributionally robust loss of $h$ around meta-distribution $\mu$:*

$$\sup_{\mu' \in \mathcal{G}_{\varepsilon}(\mu)} \mathbb{E}_{P \sim \mu'} \left( \mathbb{E}_P \left[\ell\left(y, h(\boldsymbol{X})\right)\right] \right) \leq \widehat{U}^*(\varepsilon) + \sqrt{\frac{\log\left(\frac{K+2}{\delta}\right)}{2K}} + \frac{c_2}{K} \sum_{k \in [K]} \sqrt{\frac{\log\left(\frac{(K+2)n_k}{\varepsilon\delta}\right)}{n_k}},$$

*where $c_1$ is a universal constant and $c_2$ only depends on transportation cost $c$.*

The proof is provided in Appendix C. Similar to Theorem 5.2, Theorem 6.2 offers a robust bound on the expected loss under Wasserstein meta-distributional shifts using $\widehat{U}^*(\varepsilon)$. Once again, the generalization gap decreases asymptotically as both $K$ and $\left(\min_{k \in [K]} n_k\right)/\log K$ increase. Specifically, we have:

$$\sup_{\mu' \in \mathcal{G}_{\varepsilon}(\mu)} \mathbb{E}_{\mu'} \left[\mathbb{E}_P\left(\ell\left(y, h(\boldsymbol{X})\right)\right)\right] - \widehat{U}^*(\varepsilon) \stackrel{\text{w.h.p.}}{\leq} \mathcal{O}\left(\sqrt{\frac{\log K}{K}} + \max_{k \in [K]} \sqrt{\frac{\log(Kn_k)}{n_k}}\right). \tag{13}$$

It should be noted that the inherent robustness of $h$ against Wasserstein distributional shifts, if it exists, is reflected in $\widehat{U}^*(\varepsilon)$, thereby reducing the bound. Also, once again $\widehat{U}^*(\varepsilon)$ is asymptotically minimax optimal since when both network size and client sample size go to infinity, it converges to the true (statistical) robust risk due to the strong law of large numbers. This means our bounds are tight. In the following, we also discuss the computational complexity and privacy aspects of our scheme in Remarks 6.3 and 6.4, respectively.

*Remark* 6.3 (**Computational Complexity**)**. Server-side**: The program in equation 12 is quasi-convex. Without any additional requirements or assumptions, a standard quasi-convex optimization algorithm, such as the bisection method described in Algorithm 1 (Appendix D) can approximate $\widehat{U}^*(\varepsilon)$ within an arbitrary error margin $\Delta > 0$, with polynomial time complexity relative to $K$ and $\log \frac{1}{\Delta}$. The total query budget per client is also polynomial with respect to both $K$ and $\log \frac{1}{\Delta}$. Proof of Remark 6.3 together with Algorithm 1 can be found in Appendix D. **Client-side**: Assume the transportation cost $c$ in Theorem 6.2 is convex with respect to its second argument[1] and is differentiable. Also, assume the robustness radius $\rho$ is not chosen to be excessively large. Under these conditions, the client-side optimization problem to determine the query value $\widehat{\mathsf{QV}}_k(h, \rho)$ in equation 12, for any given $h$ and $\rho$, is convex (Sinha et al., 2018). A standard stochastic gradient descent

---

[1]This assumption is generally not restrictive, as any norm exhibits this property

algorithm can approximate $\widehat{\mathrm{QV}}_k(h, \rho)$ within an arbitrary error margin $\Delta > 0$, with polynomial time complexity relative to $K$, $\Delta^{-1}$ and $\varepsilon^{-1}$.

The proof of latter statements relies on the concavity of the objective function in the adversarial loss value $\widehat{\mathrm{QV}}_k(h, \rho)$ when $\rho$ is not excessively large. Notably, this property holds regardless of whether $h$ is convex or not. In Section 2 of (Sinha et al., 2018), a detailed analysis of the convergence rate for such problems is provided, assuming that the transportation cost $c(\cdot, \cdot)$ is strongly convex with respect to one of its arguments. Such transportation costs are widely used in real-world applications and cover a broad range of options, including almost all norms.

*Remark* 6.4 (**Certificate of Privacy**). The program in equation 12 relies solely on the Wasserstein adversarial loss values from each client $k$, derived using the query value $\widehat{\mathrm{QV}}_k(h, \rho)$ for polynomially many values of $\rho$. No other information or direct access to private client data is required. To the best of our knowledge, there are no known privacy attacks capable of effectively recovering private data from this procedure.

## 7 EXPERIMENTAL RESULTS

In this section, we present a series of experiments on real-world datasets to demonstrate both: (i) the validity and tightness of our bounds, and (ii) that our bounds are not only theoretically sound but also computable in practice.

### 7.1 CLIENT GENERATION AND RISK CDF CERTIFICATES FOR UNSEEN CLIENTS

In the first part of our experiments, we outline our client generation model and present a number of risk CDF guarantees. We simulated a federated learning scenario with $n = 500$ nodes, where each node contains 1000 local samples. The experiments were conducted using four different datasets: CIFAR-10 (Krizhevsky et al., 2009), SVHN (Netzer et al., 2011), EMNIST (Cohen et al., 2017), and ImageNet (Russakovsky et al., 2015). To create each user's data within the network, we applied three types of affine distribution shifts across users:

**Feature Distribution Shift:** Each sample $\boldsymbol{X}_i^{(k)}$ in the dataset is manipulated via a transformation chosen randomly for each node. Specifically, each user is assigned a unique matrix $\Lambda^{(k)}$ and shift vector $\boldsymbol{\delta}^{(k)}$, and the data is modified as follows:

$$\tilde{\boldsymbol{X}}_i^{(k)} = (I + \Lambda^{(k)})\boldsymbol{X}_i^{(k)} + \boldsymbol{\delta}^{(k)}. \tag{14}$$

In our experiments, $\Lambda^{(k)}$ and $\boldsymbol{\delta}^{(k)}$ are respectively random matrices and vectors with i.i.d. zero-mean Gaussian entries. The standard deviation varies based on the dataset: 0.05 for CIFAR-10 and SVHN, 0.1 for EMNIST, and 0.01 for ImageNet.

**Label Distribution Shift:** Here, we assume that each meta-distribution is characterized by a specific $\alpha$ coefficient. To generate each user's data under this shift, the number of samples per class is determined by a Dirichlet distribution with parameter $\alpha$. In our experiments, we use $\alpha = 0.4$.

**Feature & Label Distribution Shift:** As the name suggests, this shift combines both the feature and label distribution shifts described above to create a new distribution for each user.

Figure 1 illustrates our performance certificates (i.e., bounds on the risk CDF) for unseen clients when there are no shifts. We selected 100 nodes from the population and considered 400 other nodes as unseen clients. We then plotted the CDFs based on 100 samples and confirmed that our bounds hold for the real population as well. Due to the standard DKW inequality, the empirical CDF is a good estimate for the test-time non-robust risk CDF.

### 7.2 CERTIFICATES FOR $f$-DIVERGENCE META-DISTRIBUTIONAL SHIFTS

In this section, we examined scenarios where users belong to two distinct meta-distributions: the source and the target. A DNN-based model is initially trained on a network of clients sampled from the source. The resulting risk values are then fed into the optimization problems introduced in Section 5 to obtain robust CDF bounds, considering both the Chi-Square and KL divergence as potential choices for $f$. Finally, we empirically estimate the risk CDF for users from the target

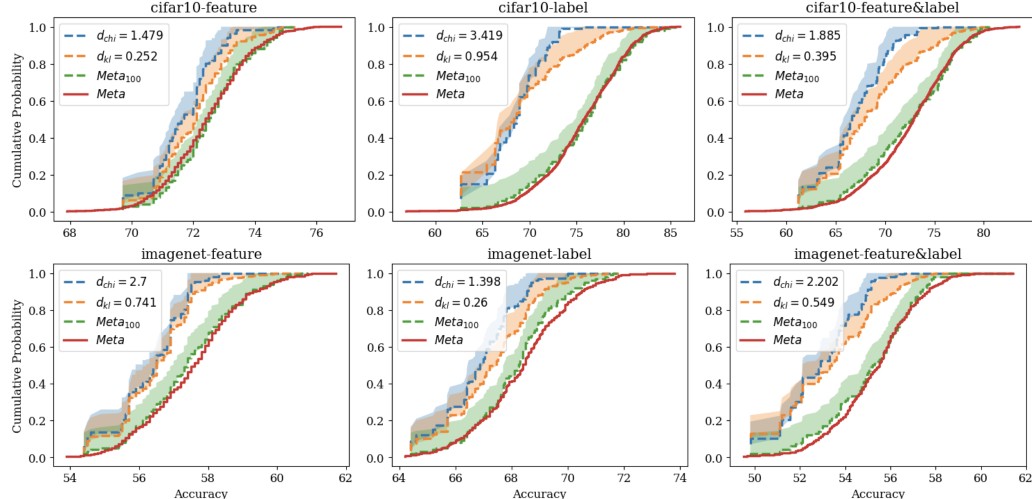

Figure 1: Non-robust Risk CDF bounds for unseen clients. Here, "*Meta*" refers to the main population with $500$ nodes. DKW-robust bounds are depicted only for tightness comparison.

meta-distribution and validate our bounds. Specifically, we tested our certificates in two distinct settings using the CIFAR-10 dataset (see Figure 2). We generated various image categories with differing resolutions or color schemes, and then sampled from these categories to create different distributions:

**Resolutions**: Images were cropped and resized to create eight different resolutions. The Dirichlet $\alpha$ coefficients for the first (source) meta-distribution range from $0.4$ to $0.7$ for the four lower resolutions and from $0.7$ to $1$ for the four higher resolutions. For the second (target) meta-distribution, the ranges are reversed: $0.7$ to $1$ for the lower resolutions and $0.4$ to $0.7$ for the higher resolutions. The number of samples per resolution for each user is determined using a Dirichlet distribution, with $\alpha$ coefficients randomly selected from the specified range for the meta-distribution. As a result, users sampled from source will have more high-resolution images, while users from the target will have more low-resolution samples.

**Colors**: The color intensity of the images varies from $0.00$ (gray-scale) to $1.00$ (fully colored). For the source meta-distribution, the $\alpha$ coefficients range from $0$ to $0.5$ for images with color intensity below $0.5$, and from $0.5$ to $1$ for images above $0.5$. As with the resolution setting, the ranges are reversed for the target, and the number of samples per color intensity for each user is calculated similarly. Therefore, users sampled from source will have more colorful images, while those from the target will have more gray-scale images.

Figure 3 (left) verifies our CDF certificates based on both chi-square and KL-divergence (dotted curves) for the target meta distribution (orange curve). As can be seen, bounds have tightly captured the behavior of risk CDF in the target network. More detailed experiments are shown in Figure 5 in Appendix E.

## 7.3 CERTIFICATES FOR WASSERSTEIN-BASED META-DISTRIBUTIONAL SHIFTS

In this experiment, as previously mentioned, we used affine distribution shifts to create new domains. Figure 3 (right) summarizes our numerical results in this scenario. To generate different networks within the meta-distribution, we applied the affine distribution shifts described in Section 7.1. Once again, the results validate our certificates, this time for Wasserstein-type shifts. The blue curve, representing the real population, consistently falls within or beneath the blue shaded area. Regarding tightness, it is important to note that the bounds presented here remain tight, particularly under adversarial attacks as defined by a distributional adversary in (Sinha et al., 2018). More detailed experiments with various levels of tightness are shown in Figure 6 in Appendix E.

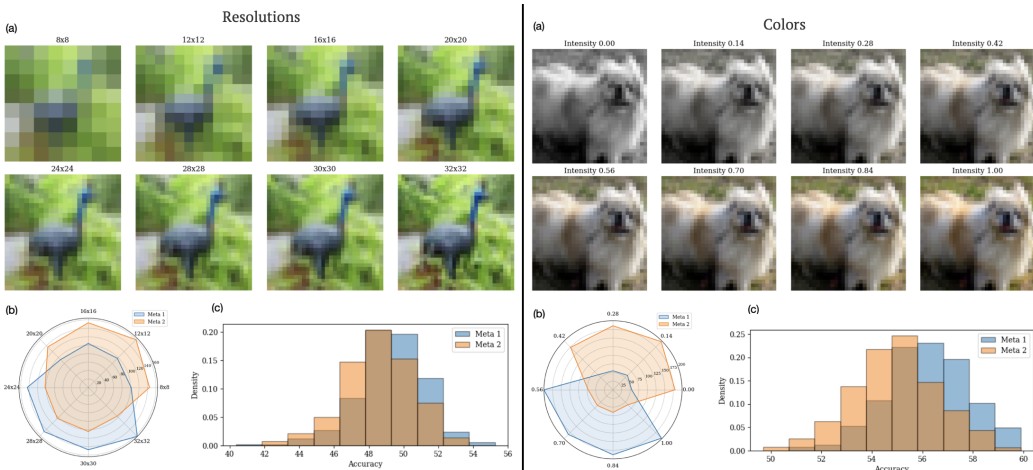

Figure 2: Meta-distributional shifts based on resolutions (left) and colors (right). (a) Sample images from each resolution/color. (b) Average number of samples per resolution/color within each network selected from the meta distributions. (c) Histogram of model accuracy densities for the two meta distributions.

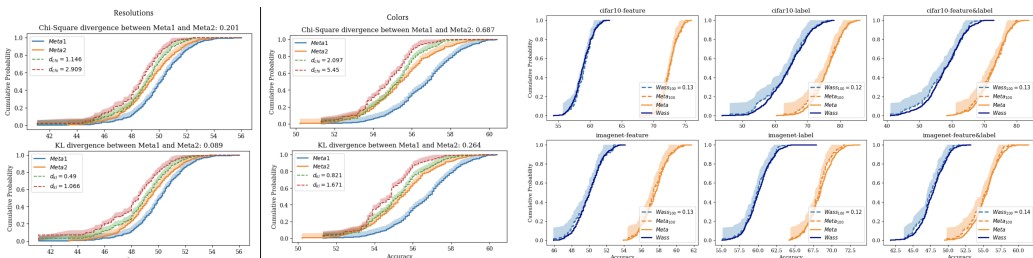

Figure 3: (Left) f-divergence certificates for two meta-distributions based on resolutions and colors. (Right) Wasserstein-based certificates for unseen clients. "*Meta*" and "*Wass*" refer to the main population with 500 nodes. Dotted curves are based on 100 networks within the population.

Although our theoretical findings in Section 6 focus solely on the average risk and not the risk CDF, we extended the same framework to the CDF in this experiment to explore whether the theory might also apply. The results were positive, suggesting potential for extending our theoretical findings in this area.

## 8 CONCLUSIONS

In this work, we proposed new performance bounds, computable in polynomial time with polynomial query complexity, to establish guarantees for any model $h$ on an unseen network B, based on observations and queries from network A. The key assumption is that the meta-distributions of both networks' user data are $\varepsilon$-close, for some known or assumed $\varepsilon > 0$. We considered two measures of "closeness": $f$-divergence shifts and Wasserstein shifts, which capture different practical scenarios, making our approach widely applicable.

Our bounds are supported by rigorous proofs, with vanishing generalization gaps, ensuring consistency as the network size $K$ and sample per client $n_k / \log K$ increase asymptotically. We also introduced a robust version of the GC theorem and DKW bound for $f$-divergence shifts, which is novel. Numerical experiments demonstrate the bounds' practical tightness and computability, as the proposed optimization problems are convex or quasi-convex, solvable in polynomial time. Future work could explore uniform convergence bounds for risk CDF under Wasserstein shifts, or consider other shift types beyond $f$-divergence and Wasserstein.

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

## A   AUXILIARY DEFINITIONS AND COROLLARIES

**Definition A.1** ($f$-Divergence Between Meta-Distributions). Consider two meta-distributions $\mu, \mu' \in \mathcal{M}^2$ where $\mu'$ is absolutely continuous with respect to $\mu$. Let $f : [0, \infty) \to [-\infty, \infty]$ be a convex function such that $f(x)$ is finite for all $x > 0$, $f(1) = 0$, and $f(0) = \lim_{t \to 0^+} f(t)$ (which could be infinite). The $f$-divergence between $\mu$ and $\mu'$ is defined as:

$$\mathcal{D}_f(\mu' \| \mu) = \int_{P \in \mathcal{M}} f\left(\frac{\mathrm{d}\mu'(P)}{\mathrm{d}\mu(P)}\right) \mathrm{d}\mu(P). \tag{15}$$

**Assumption A.2** (Density Ratio Boundedness). *Assume two meta-distributions $\mu, \mu' \in \mathcal{M}^2$ where both $\mu'$ and $\mu$ are absolutely continuous with respect to each other. For $\Lambda \geq 1$, we say $\mu$ and $\mu'$ have a $\Lambda$-bounded density ratio if the following condition holds for all $P \in \mathcal{M}$:*

$$\Lambda^{-1} \leq \inf_P \frac{\mathrm{d}\mu'(P)}{\mathrm{d}\mu(P)} \leq \sup_P \frac{\mathrm{d}\mu'(P)}{\mathrm{d}\mu(P)} \leq \Lambda. \tag{16}$$

**Definition A.3** (Wasserstein metric on $\mathcal{M}(\mathcal{Z})$). For any two measures $P, Q \in \mathcal{M}(\mathcal{Z})$ and a lower semi-continuous function $c : \mathcal{Z} \times \mathcal{Z} \to \mathbb{R}_{\geq 0}$, we define the *Wasserstein* distance between $P$ and $Q$ as

$$\mathcal{W}_c(P, Q) \triangleq \inf_{\nu \in \mathcal{C}(P,Q)} \mathbb{E}_{(\boldsymbol{Z}, \boldsymbol{Z}') \sim \nu} \{c(\boldsymbol{Z}, \boldsymbol{Z}')\}, \tag{17}$$

where $\mathcal{C}(P, Q)$ denotes the set of all couplings between $P$ and $Q$, i.e., all probability measures in $\mathcal{M}(\mathcal{Z} \times \mathcal{Z})$ that have fixed $P$ and $Q$ as their respective marginals.

The function $c$ is called the *transportation cost* and in practice, for example in a classification task with $\mathcal{Z} = \mathcal{X} \times \mathcal{Y}$, can be chosen as $c(\boldsymbol{Z}, \boldsymbol{Z}') \triangleq \|\boldsymbol{X} - \boldsymbol{X}'\|_p^q + \lambda \mathbb{1}(y \neq y')$, for any $p, q \geq 1$, $\lambda \geq 0$. However, $c$ is user-defined and can be chosen differently.

**Corollary A.4** (Robust Version of Glivenko-Cantelli Theorem). *Let $\mu$ and $\mu'$ be two probability measures on $\mathbb{R}$ and let $\mathscr{S} \triangleq \{X_i\}_{i=1}^{\infty}$ be an i.i.d. sequence drawn from $\mu$. Assume $\mu$ and $\mu'$ are absolutely continuous with respect to each other. Additionally, suppose $\mathcal{D}_f(\mu' \| \mu) \leq \varepsilon$ for some $\varepsilon \geq 0$ and a proper convex function $f(\cdot)$. Then, there exists a non-negative sequence $\{\alpha_1\}_{i=1}^{\infty}$ that can depend on $\mathscr{S}$, has the following properties:*

$$\lim_{n \to \infty} \frac{1}{n} \sum_{i=1}^{n} \alpha_i = 1 \quad \text{and} \quad \lim_{n \to \infty} \frac{1}{n} \sum_{i=1}^{n} f(\alpha_i) \leq \varepsilon, \tag{18}$$

*and also satisfies the following condition:*

$$\lim_{n \to \infty} \sup_{\lambda \in \mathbb{R}} \left| \mu'(X \leq \lambda) - \frac{1}{n} \sum_{i=1}^{n} \alpha_i \mathbb{1}(X_i \leq \lambda) \right| \stackrel{a.s.}{=} 0. \tag{19}$$

Corollary A.4 follows directly from Lemma 5.4.

# B PROOFS OF THE STATEMENTS IN SECTION 5

*Proof of Theorem 5.2.* For each $k \in [K]$, let us define the event $\xi_1^{(k)}$ as follows:

$$\xi_1^{(k)} \equiv \mathbb{E}_{P_k}[\ell(y, h(\boldsymbol{X}))] \leq \frac{1}{n_k} \sum_{i \in [n_k]} \ell\left(y_i^{(k)}, h\left(\boldsymbol{X}_i^{(k)}\right)\right) + \sqrt{\frac{\log\left(\frac{K+3}{\delta}\right)}{2n_k}}, \tag{20}$$

where, since $\ell(\cdot)$ is assumed to be a 1-bounded loss function and the samples are drawn independently, McDiarmid's inequality tells us that $\mathbb{P}\left(\xi_1^{(k)}\right) \geq 1 - \frac{\delta}{K+3}$ (McDiarmid et al., 1989). We now state and prove two essential lemmas which will be used in the subsequent arguments.

**Lemma B.1.** *Consider two meta-distributions $\mu, \mu' \in \mathcal{M}^2$ which are absolutely continuous with respect to each other and have a $\Lambda$-bounded density ratio for some $\Lambda \geq 1$. For $K \in \mathbb{N}$, assume $P_1, \ldots, P_K \in \mathcal{M}$ to be i.i.d. sample distributions sampled from $\mu$. Then, for all $\epsilon > 0$, the following concentration bound holds:*

$$\mathbb{P}\left( \left| \frac{1}{K} \sum_{k \in [K]} \frac{\mathrm{d}\mu'(P_k)}{\mathrm{d}\mu(P_k)} - 1 \right| \geq \epsilon \right) \leq \exp\left( \frac{-2K\epsilon^2}{\Lambda^2 (1 - \Lambda^{-2})^2} \right). \tag{21}$$

*Proof of Lemma B.1.* Due to the assumed mutual absolute continuity, $\mu$ and $\mu'$ share the same support. Therefore, for $P \sim \mu$, we can define the scalar random variable

$$\zeta = \zeta(P) \triangleq \frac{\mathrm{d}\mu'(P)}{\mathrm{d}\mu(P)}. \tag{22}$$

This variable is bounded by $\Lambda^{-1} \overset{a.s.}{\leq} \zeta \overset{a.s.}{\leq} \Lambda$. Regarding the expected value of $\zeta$, we have:

$$\mathbb{E}\left[\zeta\right] = \mathbb{E}_{P \sim \mu}\left[\frac{\mathrm{d}\mu'(P)}{\mathrm{d}\mu(P)}\right] = \int_{\mathcal{M}} \frac{\mathrm{d}\mu'(P)}{\mathrm{d}\mu(P)}\mathrm{d}\mu(P) = 1. \tag{23}$$

Let $\zeta_k = \zeta(P_k)$. Since $\zeta(P_1), \ldots, \zeta(P_K)$ represent i.i.d. instances of $\zeta$, McDiarmid's inequality states that:

$$\mathbb{P}\left(\left|\frac{1}{K}\sum_{k \in [K]} \zeta_k - \mathbb{E}\left[\zeta\right]\right| \geq \epsilon\right) \leq \exp\left(-\frac{2K\epsilon^2}{\left(\Lambda - \Lambda^{-1}\right)^2}\right), \tag{24}$$

which completes the proof. $\qquad\square$

**Lemma B.2.** *For $\Lambda \geq 1$, assume two meta-distributions $\mu, \mu' \in \mathcal{M}^2$ are absolutely continuous with respect to each other and have a $\Lambda$-bounded density ratio. Let $f$ be a convex function that satisfies the conditions described in Definition A.1. For $K \in \mathbb{N}$, assume $P_1, \ldots, P_K \in \mathcal{M}$ to be i.i.d. sample distributions sampled from $\mu$. Then, the following concentration bound holds:*

$$\mathbb{P}\left(\left|\mathcal{D}_f\left(\mu'\|\mu\right) - \frac{1}{K}\sum_{k \in [K]} f\left(\frac{\mathrm{d}\mu'\left(P_k\right)}{\mathrm{d}\mu\left(P_k\right)}\right)\right| \geq \epsilon\right) \leq \exp\left(\frac{-2K\epsilon^2}{\mathsf{BW}^2\left(f\left(\cdot\right), \Lambda\right)}\right), \tag{25}$$

*where* $\mathsf{BW}\left(f\left(\cdot\right), \Lambda\right)$ *is defined as:*

$$\mathsf{BW}\left(f\left(\cdot\right), \Lambda\right) \triangleq \sup_{\Lambda^{-1} \leq u, v \leq \Lambda} f(u) - f(v). \tag{26}$$

*Proof of Lemma B.2.* The proof follows similarly to that of Lemma B.1. For $P \sim \mu$, let us define the random variable

$$\zeta(P) = f\left(\frac{\mathrm{d}\mu'\left(P\right)}{\mathrm{d}\mu\left(P\right)}\right). \tag{27}$$

Then, having defined $\zeta_k = \zeta\left(P_k\right)$, we know that $\zeta_1, \ldots, \zeta_K$ represent i.i.d. instances of $\zeta$. Moreover, the expected value of $\zeta$ is the $f$-divergence between $\mu'$ and $\mu$:

$$\mathbb{E}\left[\zeta\right] = \mathbb{E}_{P \sim \mu}\left[f\left(\frac{\mathrm{d}\mu'(P)}{\mathrm{d}\mu(P)}\right)\right] = \mathcal{D}_f\left(\mu'\|\mu\right). \tag{28}$$

Finally, since $\mu'$ and $\mu$ have the $\Lambda$-bounded density ratio property, the following bounds hold almost surely:

$$\zeta \overset{a.s.}{\leq} \sup_{\Lambda^{-1} \leq u \leq \Lambda} f(u) \quad, \quad \zeta \overset{a.s.}{\geq} \inf_{\Lambda^{-1} \leq v \leq \Lambda} f(v), \tag{29}$$

which means the range of $\zeta$ is almost surely equal to $\mathsf{BW}\left(f\left(\cdot\right), \Lambda\right)$. Hence, again using McDiarmid's inequality, we get the bound:

$$\mathbb{P}\left(\left|\frac{1}{K}\sum_{k \in [K]} \zeta_k - \mathbb{E}\left[\zeta\right]\right| \geq \epsilon\right) \leq \exp\left(\frac{-2K\epsilon^2}{\mathsf{BW}^2\left(f\left(\cdot\right), \Lambda\right)}\right), \tag{30}$$

and this completes the proof. $\qquad\square$

With the lemmas established, let us define additional events $\xi_2$ and $\xi_3$ based on the concentration bounds:

$$\xi_2 \equiv \frac{1}{K}\sum_{k \in [K]} \frac{\mathrm{d}\mu'(P_k)}{\mathrm{d}\mu(P_k)} \overset{<}{\underset{>}{\lessgtr}} 1 \pm C_1 K^{-1/2}, \tag{31}$$

$$\xi_3 \equiv \frac{1}{K}\sum_{k \in [K]} f\left(\frac{\mathrm{d}\mu'(P_k)}{\mathrm{d}\mu(P_k)}\right) \overset{<}{\underset{>}{\lessgtr}} \mathcal{D}_f\left(\mu'\|\mu\right) \pm C_2 K^{-1/2}, \tag{32}$$

where $C_1, C_2$ are constants which only depend on $\Lambda$ and $f(\cdot)$, according to Lemmas B.1 and $B.2$. Based on the above arguments and the results of the mentioned lemmas, we have $\mathbb{P}(\xi_2), \mathbb{P}(\xi_2) \geq 1 - \frac{\delta}{K+3}$. Using the central idea for *importance sampling* (Glynn & Iglehart, 1989), the following equations hold for all $\mu, \mu', \ell$ and $h$:

$$\mathbb{E}_{P \sim \mu} \left[ \left( \frac{\mathrm{d}\mu'(P)}{\mathrm{d}\mu(P)} \right) \mathbb{E}_P \left[ \ell(y, h(\boldsymbol{X})) \right] \right] = \int_{P \in \mathcal{M}} \left( \frac{\mathrm{d}\mu'(P)}{\mathrm{d}\mu(P)} \right) \mathbb{E}_P \left[ \ell(y, h(\boldsymbol{X})) \right] \mathrm{d}\mu(P)$$
$$= \mathbb{E}_{P \sim \mu'} \left[ \mathbb{E}_P \left[ \ell(y, h(\boldsymbol{X})) \right] \right]. \quad (33)$$

At this point, and similar to the idea of Lemma B.1, we define $\xi_4$ as the event of the empirical loss over meta-distribution $\mu'$ concentrates (with high probability) around its expected value, i.e.,

$$\xi_4 \equiv$$

$$\mathbb{E}_{P \sim \mu} \left[ \left( \frac{\mathrm{d}\mu'(P)}{\mathrm{d}\mu(P)} \right) \mathbb{E}_P \left[ \ell(y, h(\boldsymbol{X})) \right] \right] \leq \frac{1}{K} \sum_{k \in [K]} \frac{\mathrm{d}\mu'(P_k)}{\mathrm{d}\mu(P_k)} \mathbb{E}_{P_k} \left[ \ell(y, h(\boldsymbol{X})) \right] + \Lambda \sqrt{\frac{\log\left(\frac{K+3}{\delta}\right)}{2K}}. \quad (34)$$

Again, since

$$0 \overset{a.s.}{\leq} \frac{\mathrm{d}\mu'(P_k)}{\mathrm{d}\mu(P_k)} \mathbb{E}_{P_k} \left[ \ell(y, h(\boldsymbol{X})) \right] \overset{a.s.}{\leq} \Lambda,$$

McDiarmid's inequality states that the probability bound $\mathbb{P}(\xi_4) \geq 1 - \frac{\delta}{K+3}$ holds. Our final definition in this proof is a random set of meta-distributions $\mathcal{G} \subseteq \mathcal{M}^2$ which represents an empirical candidate for the neighbors of $\mu$. Mathematically speaking, let us define:

$$\mathcal{G} \triangleq \left\{ \nu \in \mathcal{M}^2 \,\middle|\, \frac{1}{K} \sum_{k \in [K]} \frac{\mathrm{d}\nu(P_k)}{\mathrm{d}\mu(P_k)} \overset{\leq}{\underset{\geq}{\lessgtr}} 1 \pm C_1 K^{-1/2} \,,\, \frac{1}{K} \sum_{k \in [K]} f\left( \frac{\mathrm{d}\nu(P_k)}{\mathrm{d}\mu(P_k)} \right) \leq \varepsilon + C_2 K^{-1/2} \right\}, \quad (35)$$

which depends on $\varepsilon$ and has a random (empirical) nature since it also depends on sample distributions $P_1, \ldots, P_K$. Based on prior discussions and lemmas, we have $\mu' \overset{a.s.}{\in} \mathcal{G}$ as long as the events $\xi_2$ and $\xi_3$ hold, simultaneously. By further assuming that events $\xi_4$ and $\xi_1^{(k)}$s for all $k \in [K]$ also hold, we can finally write the following chain of inequalities:

$$\mathbb{E}_{P \sim \mu'} \left[ \mathbb{E}_P \left[ \ell(y, h(\boldsymbol{X})) \right] \right]$$

$$\leq \frac{1}{K} \sum_{k \in [K]} \frac{\mathrm{d}\mu'(P_k)}{\mathrm{d}\mu(P_k)} \mathbb{E}_{P_k} \left[ \ell(y, h(\boldsymbol{X})) \right] + \Lambda \sqrt{\frac{\log\left(\frac{K+3}{\delta}\right)}{2K}}$$

$$\leq \frac{1}{K} \sum_{k \in [K]} \frac{\mathrm{d}\mu'(P_k)}{\mathrm{d}\mu(P_k)} \widehat{\mathbb{E}}_{P_k} \left[ \ell(y, h(\boldsymbol{X})) \right] + \Lambda \sqrt{\frac{\log\left(\frac{K+3}{\delta}\right)}{2K}} + \frac{1}{K} \sum_{k \in [K]} \sqrt{\frac{\log\left(\frac{K+3}{\delta}\right)}{2n_k}}$$

$$\overset{a.s.}{\leq} \frac{1}{K} \sup_{\nu \in \mathcal{G}} \sum_{k \in [K]} \frac{\mathrm{d}\nu(P_k)}{\mathrm{d}\mu(P_k)} \widehat{\mathbb{E}}_{P_k} \left[ \ell(y, h(\boldsymbol{X})) \right] + \sqrt{\log\left(\frac{K+3}{\delta}\right)} \left[ \sqrt{\frac{\Lambda^2}{2K}} + \frac{1}{K} \sum_{k \in [K]} \sqrt{\frac{1}{2n_k}} \right]. \quad (36)$$

It should be noted that the condition $\nu \in \mathcal{G}$ can be interpreted as introducing

$$\alpha_k \triangleq \frac{\mathrm{d}\nu(P_k)}{\mathrm{d}\mu(P_k)}, \quad \forall k \in [K],$$

and force $\alpha_1 \ldots, \alpha_K$ to satisfy the constraints in the definition of $\widehat{B}^*(\varepsilon)$. Hence, this gives us the high probability bound claimed inside the statement of theorem. The only remaining part of the proof is to show events $\xi_1^{(k)}, \xi_2, \xi_3$ and $\xi_4$ for all $k \in [K]$ hold, simultaneously, with a probability at least $1 - \delta$.

For any event $\xi$, let $\xi^c$ denote its complement. Then, we already have

$$\mathbb{P}\left(\xi_1^{(k)c}\right), \mathbb{P}\left(\xi_2^c\right), \dots, \mathbb{P}\left(\xi_4^c\right) \le \frac{\delta}{K+3}, \quad \forall k \in [K].$$

In this regard, one can simply use the union bound and obtain the following chain of inequalities:

$$\mathbb{P}\left(\bigcup_{k \in [K]} \xi_1^{(k)c} \cup \xi_2 \cup \xi_3 \cup \xi_4\right) \le \sum_{k \in [K]} \mathbb{P}\left(\xi_1^{(k)c}\right) + \sum_{i=2}^{4} \mathbb{P}\left(\xi_i^c\right) = \frac{K\delta}{K+3} + 3\frac{\delta}{K+3} = \delta. \quad (37)$$

This means the bound in the statement of theorem holds with a probability at least $1 - \delta$, and thus completes the proof. $\qquad\square$

*Proof of Lemma 5.4.* The proof for most of its initial parts follows the same path as in the proof of Theorem 5.2. In particular, we use Lemmas B.1 and B.2 from the proof of Theorem 5.2 to show that the following events occur separately with probability at least $1 - \frac{\delta}{3}$, for any $\delta > 0$:

$$\left|\frac{1}{n}\sum_{i \in [n]} \frac{\mathrm{d}\mu'(X_i)}{\mathrm{d}\mu(X_i)} - 1\right| \le c_1 \sqrt{\frac{\log \frac{1}{\delta}}{n}}, \quad (38)$$

$$\left|\mathcal{D}_f\left(\mu' \| \mu\right) - \frac{1}{n}\sum_{i \in [n]} f\left(\frac{\mathrm{d}\mu'(X_i)}{\mathrm{d}\mu(X_i)}\right)\right| \le c_2 \sqrt{\frac{\log \frac{1}{\delta}}{n}}, \quad (39)$$

where $c_1, c_2 > 0$ are constants depending on $f(\cdot)$. These probabilities are with respect to the randomness in drawing i.i.d. samples $X_1, \dots, X_n \sim \mu$. This is equivalent to the following statement:

$$\mathbb{P}\left[\left(\frac{\mathrm{d}\mu'(X_i)}{\mathrm{d}\mu(X_i)}\right)_{i \in [n]} \in \mathcal{A}_n\right] \ge 1 - \frac{2\delta}{3}. \quad (40)$$

Next, define $\widehat{F}_n(\lambda)$ for $\lambda \in \mathbb{R}$ as

$$\widehat{F}_n(\lambda) \triangleq \frac{1}{n}\sum_{i=1}^{n} \omega(X_i)\mathbb{1}(X_i \le \lambda), \quad (41)$$

where the weight function $\omega(X) \in \left[\Lambda^{-1}, \Lambda\right]$ is the unknown (bounded) density ratio between $\mu'$ and $\mu$, i.e.,

$$\omega(X) \triangleq \frac{\mathrm{d}\mu'(X)}{\mathrm{d}\mu(X)}. \quad (42)$$

Since $\omega(\cdot)$ is non-negative, $\widehat{F}_n(\lambda)$ is non-decreasing in $\lambda$, starting at 0 when $\lambda = -\infty$ and not exceeding $\Lambda$ as $\lambda \to \infty$.

Consider the probability measure $\mu'$. For $m \ge 2$, define $\lambda_0^*, \lambda_1^*, \dots, \lambda_m^*$ such that (i) $\lambda_0^* = -\infty$ and $\lambda_m^* = \infty$, and (ii) $\mu'(X \le \lambda_i^*) = i/m$ for $i \in [m-1]$. These $\lambda_i^*, i \in [m] \cup \{0\}$ represent the $m$-quantiles of $\mu'$. For any $\lambda \in \mathbb{R}$, let $i = i(\lambda) \in [m]$ be such that $\lambda \in [\lambda_{i-1}^*, \lambda_i^*)$. Then, the following chain of inequalities holds almost surely for all $\lambda \in \mathbb{R}$:

$$\widehat{F}_n(\lambda) - \mu'(X \le \lambda) \le \widehat{F}_n(\lambda_i^*) - \mu'(X \le \lambda_{i-1}^*) \le \widehat{F}_n(\lambda_i^*) - \mu'(X \le \lambda_i^*) + \frac{1}{m},$$

$$\widehat{F}_n(\lambda) - \mu'(X \le \lambda) \ge \widehat{F}_n(\lambda_{i-1}^*) - \mu'(X \le \lambda_i^*) \ge \widehat{F}_n(\lambda_{i-1}^*) - \mu'(X \le \lambda_{i-1}^*) - \frac{1}{m}. \quad (43)$$

Thus, the following bound holds for all $\lambda \in \mathbb{R}$:

$$\left\|\widehat{F}_n - \mu'(X \le \cdot)\right\|_\infty = \sup_{\lambda \in \mathbb{R}} \left|\widehat{F}_n(\lambda) - \mu'(X \le \lambda)\right| \le \max_{i \in \{0,1,\dots,m\}} \left|\widehat{F}_n(\lambda_i^*) - \mu'(X \le \lambda_i^*)\right| + \frac{1}{m}. \quad (44)$$

On the other hand, for any fixed $\lambda \in \mathbb{R}$, we have the following relation for the expectation of $\widehat{F}_n(\lambda)$:

$$
\begin{aligned}
\mathbb{E}_\mu[\widehat{F}_n(\lambda)] &= \mathbb{E}_\mu\left[\frac{1}{n}\sum_{i=1}^n \omega(X_i)\mathbb{1}(X_i \le \lambda)\right] \\
&= \frac{1}{n}\sum_{i=1}^n \mathbb{E}_\mu\left[\omega(X_i)\mathbb{1}(X_i \le \lambda)\right] \\
&= \frac{1}{n}\sum_{i=1}^n \int_{\mathbb{R}} \frac{\mathrm{d}\mu'(X)}{\mathrm{d}\mu(X)}\mathbb{1}(X \le \lambda)\mathrm{d}\mu(X) \\
&= \mu'(X \le \lambda).
\end{aligned}
\tag{45}
$$

Given that the weight functions $\omega(X_i)$ for $i \in [n]$ are bounded in the interval $\left[\Lambda^{-1}, \Lambda\right]$, and $\mathbb{1}(\cdot) \in \{0,1\}$, McDiarmid's inequality states that for any $\varepsilon > 0$,

$$
\mathbb{P}\left(\left|\widehat{F}_n(\lambda_i^*) - \mu'(X \le \lambda_i^*)\right| > \varepsilon\right) \le 2e^{-2n\varepsilon^2/\Lambda^2}.
\tag{46}
$$

Therefore, using the union bound over all $i = 0, 1, \ldots, m$, we obtain:

$$
\mathbb{P}\left(\max_{i\in\{0,1,\ldots,m\}}\left|\widehat{F}_n(\lambda_i^*) - \mu'(X \le \lambda_i^*)\right| > \varepsilon\right) \le 2(m+1)e^{-2n\varepsilon^2/\Lambda^2}.
\tag{47}
$$

Equivalently, for any $\delta > 0$, the following bound holds with probability at least $1 - \delta/3$:

$$
\max_{i\in\{0,1,\ldots,m\}}\left|\widehat{F}_n(\lambda_i^*) - \mu'(X \le \lambda_i^*)\right| \le \Lambda\sqrt{\frac{\log\left(\frac{6(m+1)}{\delta}\right)}{2n}}.
\tag{48}
$$

Using the preceding inequalities, in particular relations in equation 40, equation 44 and equation 48, we can say there exists $\boldsymbol{\alpha} \in \mathcal{A}_n$ such that the following bounds for $\widehat{F}_n$ hold with probability at least $1 - \delta$:

$$
\begin{aligned}
\left\|\widehat{F}_n - \mu'(X \le \cdot)\right\|_\infty &\le \max_{i\in\{0,1,\ldots,m\}}\left|\widehat{F}_n(\lambda_i^*) - \mu'(X \le \lambda_i^*)\right| + \frac{1}{m} \\
&\le \inf_{m\in\mathbb{N}_{\ge 2}}\left\{\Lambda\sqrt{\frac{\log\left(\frac{6(m+1)}{\delta}\right)}{2n}} + \frac{1}{m}\right\} \\
&\le \mathcal{O}\left(\Lambda\sqrt{\frac{\log\left(\frac{n}{\delta}\right)}{n}}\right).
\end{aligned}
\tag{49}
$$

Thus, the proof is complete. $\qquad\square$

*Proof of Theorem 5.5.* Similar to the proof of Theorem 5.2, we begin by noting that, due to McDiarmid's inequality, for any $\delta > 0$, with probability at least $1 - \frac{K\delta}{K+2}$, the following set of inequalities holds simultaneously for all $k \in [K]$:

$$
\widehat{\mathsf{QV}}(h) \le \mathsf{QV}(h) + \sqrt{\frac{\log\left(\frac{K+2}{\delta}\right)}{2n_k}}.
\tag{50}
$$

Next, it can be readily verified that:

$$
\mathbb{1}(\mathsf{QV}(h) \ge \lambda) \le \mathbb{1}\left(\widehat{\mathsf{QV}}(h) \ge \lambda - \sqrt{\frac{\log\left(\frac{K+2}{\delta}\right)}{2n_k}}\right), \quad \forall k \in [K].
\tag{51}
$$

Additionally, note that:

$$
\mathbb{E}_{P\sim\mu'}\left[\mathbb{1}(\mathsf{QV}(h) \ge \lambda)\right] = \mu'(\mathsf{QV}(h) \ge \lambda).
\tag{52}
$$

The remainder of the proof simply involves applying the result of Lemma 5.4 with a maximum error probability of $\frac{2\delta}{K+2}$. This concludes the proof. $\qquad\square$

## C Proofs of the Statements in Section 6

*Proof of Theorem 6.2.* Proof consists of two parts:

- Proving the statement of theorem for the statistical case, where $\min_{k\in[K]} n_k \to \infty$ and thus we have $\widehat{\mathsf{QV}}_k(h,\rho) = \mathsf{QV}_k(h,\rho)$ for all $h \in \mathcal{H}$, $\rho \geq 0$ and $k \in [K]$.

- Replacing the statistically exact adversarial loss $\mathsf{QV}_k(h,\rho)$ which is based on the unknown distribution sample $P_k$ with its empirically calculated counterpart $\widehat{\mathsf{QV}}_k(h,\rho)$ which is computed based on the known (yet private) distribution $\widehat{P}_k$ for all $k \in [K]$. This part of the proof requires establishing a uniform convergence bound over all values of $\rho \geq 0$.

**Part I** The core mathematical tool used throughout the proof is the following duality result from (Sinha et al., 2018) (originally derived in (Blanchet & Murthy, 2019)) which works for general Wasserstein-constrained optimization problems:

**Lemma C.1** (Proposition 1 of (Sinha et al., 2018)). *Let $P$ be a probability measure defined over a measurable space $\Omega$, $\ell(\cdot) : \Omega \to \mathbb{R}$ be any loss function, $c$ denote a proper and lower semi-continuous transportation cost on $\Omega \times \Omega$, and assume $\varepsilon \geq 0$. Then, the following equality holds for the Wasserstein-constrained DRO around $P$:*

$$\sup_{Q \in \mathcal{B}_{\varepsilon}^{\text{wass}}(P)} \mathbb{E}_Q\left[\ell(\boldsymbol{Z})\right] = \inf_{\gamma \geq 0} \left\{ \gamma\varepsilon + \mathbb{E}_P\left[\sup_{\boldsymbol{Z}' \in \Omega} \ell(\boldsymbol{Z}') - \gamma c(\boldsymbol{Z}', \boldsymbol{Z})\right]\right\}. \tag{53}$$

Proof can be found inside the reference. Also, (Blanchet & Murthy, 2019) and (Zhang et al., 2024) along with several other papers have theoretically analyzed alternative proofs. Based on the duality formulation in Lemma C.1, and considering the fact that meta-distribution $\mu$ is also a "distribution" over the measurable space $\mathcal{M}$, one can rewrite the original Wasserstein-constrained MDRO in the statement of the theorem in its dual form:

$$\sup_{\mu' \in \mathcal{G}_{\varepsilon}(\mu)} \mathbb{E}_{P \sim \mu'}\left[\mathbb{E}_P\left[\ell(y, h(\boldsymbol{X}))\right]\right] = \inf_{\gamma \geq 0} \left\{ \gamma\varepsilon + \mathbb{E}_{P \sim \mu}\left[\sup_Q \mathbb{E}_Q\left[\ell(y, h(\boldsymbol{X}))\right] - \gamma\mathcal{W}_c(P, Q)\right]\right\}$$

$$= \inf_{\gamma \geq 0} \left\{ \mathbb{E}_{\mu}\left[\sup_Q \mathbb{E}_Q\left[\ell(y, h(\boldsymbol{X}))\right] - \gamma\left(\mathcal{W}_c(P, Q) - \varepsilon\right)\right]\right\}. \tag{54}$$

The main advantage achieved by this reformulation is the substitution of $\mu'$ with the fixed meta-distribution $\mu$ inside the expectation operators. Therefore, the optimization no longer has to be carried out in the $\mathcal{M}^2$ space. For the sake of simplicity in the proof, assume supreme value in equation 54 is attainable. This assumption is not necessary, and can be relaxed by using a more detailed mathematical analysis which is replacing the optimal distribution $Q^*$ with a Cauchy series of distributions and proceed with similar arguments. However, we have decided to avoid this scenario in order to simplify the proof. In this regard, let us define:

$$Q^*(P, \gamma; \varepsilon) \triangleq \arg\max_Q \mathbb{E}_Q\left[\ell(y, h(\boldsymbol{X}))\right] - \gamma\left(\mathcal{W}_c(P, Q) - \varepsilon\right), \quad \forall P \in \text{supp}(\mu). \tag{55}$$

Then, the following relation holds:

$$\sup_{\mu' \in \mathcal{G}_{\varepsilon}(\mu)} \mathbb{E}_{P \sim \mu'}\left[\mathbb{E}_P\left[\ell(y, h(\boldsymbol{X}))\right]\right] = \inf_{\gamma \geq 0} \left\{ \mathbb{E}_{\mu}\left[\mathbb{E}_{Q^*(P,\gamma)}\left[\ell(y, h(\boldsymbol{X}))\right] - \gamma\left(\mathcal{W}_c(P, Q^*(P,\gamma)) - \varepsilon\right)\right]\right\}$$

$$= \inf_{\gamma \geq 0} \left\{ \mathbb{E}_{P \sim \mu}\left[\mathbb{E}_{Q^*(P,\gamma)}\left[\ell(y, h(\boldsymbol{X}))\right]\right] - \right.$$
$$\left. \gamma\mathbb{E}_{P \sim \mu}\left[\mathcal{W}_c(P, Q^*(P,\gamma)) - \varepsilon\right]\right\}. \tag{56}$$

which, can be simply rewritten as:

$$\sup_{\mu' \in \mathcal{G}_{\varepsilon}(\mu)} \mathbb{E}_{P \sim \mu'}\left[\mathbb{E}_P\left[\ell(y, h(\boldsymbol{X}))\right]\right] = \inf_{\gamma \geq 0} \mathbb{E}_{\mu}\left[\mathbb{E}_{Q^*(P,\gamma)}\left[\ell(y, h(\boldsymbol{X}))\right]\right]$$

$$\text{subject to} \quad \mathbb{E}_{P \sim \mu}\left[\mathcal{W}_c(P, Q^*(P,\gamma))\right] \leq \varepsilon. \tag{57}$$

Using a similar argument as before, let us assume the $\inf_{\gamma \geq 0}$ in equation 57 is also attainable and denote the optimal value by $\gamma^* = \gamma^* (\mu, \varepsilon)$. Once again, this assumption is not necessary and can be relaxed at the expense of introducing more mathematical details and making the proof less readable. In this regard, we have:

$$\sup_{\mu' \in \mathcal{G}_\varepsilon(\mu)} \mathbb{E}_{P \sim \mu'} \left[ \mathbb{E}_P \left[ \ell \left( y, h \left( \boldsymbol{X} \right) \right) \right] \right] = \mathbb{E}_\mu \left[ \mathbb{E}_{Q^*(P, \gamma^*)} \left[ \ell \left( y, h \left( \boldsymbol{X} \right) \right) \right] \right], \tag{58}$$

where it has been already guaranteed that the optimal parameter $\gamma^* \geq 0$ and optimal distribution $Q^* (P, \gamma^*)$, the following constraint holds:

$$\mathbb{E}_{P \sim \mu} \left[ \mathcal{W}_c \left( P, Q^* \left( P, \gamma^* \right) \right) \right] \leq \varepsilon. \tag{59}$$

For any $P \in \operatorname{supp}(\mu) \subseteq \mathcal{M}$, let us define the following *optimal robustness radius* function

$$\rho^* \left( P; \varepsilon, \mu \right) \triangleq \mathcal{W}_c \left( P, Q^* \left( P, \gamma^* \right) \right). \tag{60}$$

Therefore, the original MDRO objective in the statement of the theorem can be readily upperbounded using the following distributionally robust formulation:

$$\sup_{\mu' \in \mathcal{G}_\varepsilon(\mu)} \mathbb{E}_{P \sim \mu'} \left[ \mathbb{E}_P \left[ \ell \left( y, h \left( \boldsymbol{X} \right) \right) \right] \right] = \mathbb{E}_{P \sim \mu} \left[ \mathbb{E}_{Q^*(P, \gamma^*)} \left[ \ell \left( y, h \left( \boldsymbol{X} \right) \right) \right] \right]$$

$$\leq \mathbb{E}_{P \sim \mu} \left[ \sup_{Q \in \mathcal{B}_{\rho^*(P)}^{\mathrm{wass}}(P)} \mathbb{E}_Q \left[ \ell \left( y, h \left( \boldsymbol{X} \right) \right) \right] \right]. \tag{61}$$

Using the upper-bound in equation 61 and the inequality condition on optimal Wasserstein radius functions $\rho^*(P)$ described in equation 59, we can proceed to the empirical stage of the proof. At this stage, the true expectation operators should be replaced by their empirical counterparts which are based on i.i.d. realizations of meta-distribution $\mu$, i.e., unknown distributions $P_1, \ldots, P_K$ and their known yet private empirical realizations, i.e., $\widehat{P}_i$ for $i \in [K]$.

For $P \sim \mu$, let us define the following new and real-valued random variables $\psi(P)$ and $\zeta(P)$ as follows:

$$\psi(P) \triangleq \sup_{Q \in \mathcal{B}_{\rho^*(P)}^{\mathrm{wass}}(P)} \mathbb{E}_Q \left[ \ell \left( y, h \left( \boldsymbol{X} \right) \right) \right],$$

$$\zeta(P) \triangleq \rho^* \left( P; \varepsilon, \mu \right). \tag{62}$$

It should be noted that $\psi(P)$ is readily known to be (almost surely) bounded by 1, since $\ell(\cdot)$ is assumed to be 1-bounded. Additionally, the boundedness for $\zeta(P)$ directly results from the assumption that $c$ is a bounded transportation cost.

**Lemma C.2.** *There exists $R < +\infty$ such that We have $\rho^*(P; \mu, \varepsilon) \overset{a.s.}{<} R$ for $P \sim \mu$.*

*Proof.* The proof is straightforward and directly results from the definition of Wasserstein distance:

$$\zeta(P) \triangleq \rho^* \left( P; \varepsilon, \mu \right) = \mathcal{W}_c \left( P, Q^* \left( P, \gamma^* \right) \right)$$

$$= \inf_{\nu \in \mathcal{C}(P, Q^*)} \mathbb{E}_\nu \left[ c \left( \boldsymbol{Z}, \boldsymbol{Z}' \right) \right]$$

$$\overset{a.s.}{\leq} \sup_{\boldsymbol{Z}, \boldsymbol{Z}' \in \mathcal{Z}} c \left( \boldsymbol{Z}, \boldsymbol{Z}' \right) < +\infty, \tag{63}$$

which concludes the proof. $\square$

Using a similar series of arguments to the ones explained in Lemmas B.1 and B.2 (proof of Theorem 5.2), together with the fact that $\psi(P_k)$s are all bounded by 1, one can directly apply the McDiarmid's inequality and show that the following bound holds with probability at least $1 - \frac{\delta}{K+2}$, for any $\delta > 0$:

$$\mathbb{E}_{P \sim \mu} \left[ \sup_{Q \in \mathcal{B}_{\rho^*(P)}^{\mathrm{wass}}(P)} \mathbb{E}_Q \left[ \ell \left( y, h \left( \boldsymbol{X} \right) \right) \right] \right]$$

$$\leq \frac{1}{K} \sum_{k \in [K]} \sup_{Q \in \mathcal{B}_{\rho^*(P_k)}^{\mathrm{wass}}(P_k)} \mathbb{E}_Q \left[ \ell \left( y, h \left( \boldsymbol{X} \right) \right) \right] + \sqrt{\frac{\log \left( \frac{K+2}{\delta} \right)}{2K}}. \tag{64}$$

On the other hand, by using the boundedness property for $\zeta(P)$ proved in Lemma C.2 and applying McDiarmid's inequality once again, the following bound holds with probability $1 - \frac{\delta}{K+2}$ (for any $\delta > 0$) for the empirical mean of $\zeta(P)$ over true sample distributions $P_1, \ldots, P_k$:

$$\frac{1}{K} \sum_{k \in [K]} \zeta(P_k) \leq \mathbb{E}_{P \sim \mu}[\zeta(P)] + c_1 \sqrt{\frac{\log\left(\frac{K+2}{\delta}\right)}{K}} \leq \varepsilon + c_1 \sqrt{\frac{\log\left(\frac{K+2}{\delta}\right)}{K}}, \quad (65)$$

where $c_1$ is a known universal constant depending only on the bound on transportation cost $c$. Here, the last inequality is a direct consequence of the property shown in equation 59.

Let $\mathcal{S} \subset \mathbb{R}_{\geq 0}^K$ be defined as the following subset:

$$\mathcal{S} \triangleq \left\{ (\zeta_1, \ldots, \zeta_K) \in \mathbb{R}^K \,\middle|\, \zeta_k \geq 0, \ \forall k \in [K], \ \frac{1}{K} \sum_{k \in [K]} \zeta_k \leq \varepsilon + c_1 \sqrt{\frac{\log\left(\frac{K+2}{\delta}\right)}{K}} \right\}. \quad (66)$$

So far, we have shown that

$$\mathbb{P}\left( \{\zeta(P_k)\}_{k \in [K]} \in \mathcal{S} \right) \geq 1 - \frac{\delta}{K+2}. \quad (67)$$

In a similar procedure to the one used in the proof of Theorem 5.2, union bound ensures that the bound in equation 64 and the mathematical statement of $\{\zeta(P_k)\}_{k \in [K]} \in \mathcal{S}$ simultaneously hold with probability at least $1 - \frac{2\delta}{K+2}$. Then the following chain of bounds also hold with the same probability w.r.t. drawing of $P_1, \ldots, P_K$ from $\mu$:

$$\sup_{\mu' \in \mathcal{G}_\varepsilon(\mu)} \mathbb{E}_{P \sim \mu'} \left[ \mathbb{E}_P\left[\ell\left(y, h\left(\boldsymbol{X}\right)\right)\right] \right] \leq \mathbb{E}_{P \sim \mu} \left[ \sup_{Q \in \mathcal{B}_{\rho^*(P)}^{\text{wass}}(P)} \mathbb{E}_Q\left[\ell\left(y, h\left(\boldsymbol{X}\right)\right)\right] \right]$$

$$\leq \frac{1}{K} \sum_{k \in [K]} \sup_{Q \in \mathcal{B}_{\rho^*(P_k)}^{\text{wass}}(P_k)} \mathbb{E}_Q\left[\ell\left(y, h\left(\boldsymbol{X}\right)\right)\right] + \sqrt{\frac{\log\left(\frac{K+2}{\delta}\right)}{2K}}$$

$$\leq \sup_{\underline{\rho} \in \mathcal{S}} \frac{1}{K} \sum_{k \in [K]} \sup_{Q \in \mathcal{B}_{\rho_k}^{\text{wass}}(P_k)} \mathbb{E}_Q\left[\ell\left(y, h\left(\boldsymbol{X}\right)\right)\right] + \sqrt{\frac{\log\left(\frac{K+2}{\delta}\right)}{2K}}$$

$$= \sup_{\underline{\rho} \in \mathcal{S}} \frac{1}{K} \sum_{k \in [K]} \mathsf{QV}_k(h, \rho_k) + \sqrt{\frac{\log\left(\frac{K+2}{\delta}\right)}{2K}}. \quad (68)$$

For reasons that become clear in the final stages of the proof, we need to replace the set $\mathcal{S}$ with a new one denoted by $\mathcal{S}'$ which should be defined as:

$$\mathcal{S}' \triangleq \left\{ (\zeta_1, \ldots, \zeta_K) \in \mathbb{R}^K \,\middle|\, \zeta_k \geq \frac{\varepsilon}{K}, \ \forall k \in [K], \ \frac{1}{K} \sum_{k \in [K]} \zeta_k \leq \varepsilon\left(1 + \frac{1}{K}\right) + c_1 \sqrt{\frac{\log\left(\frac{K+2}{\delta}\right)}{2K}} \right\}. \quad (69)$$

Evidently, replacing $\mathcal{S}'$ with $\mathcal{S}$ in the maximization step of equation 68, i.e., $\sup_{\underline{\rho} \in \mathcal{S}'}$, gives an upper bound for the original formulation of $\sup_{\underline{\rho} \in \mathcal{S}}$, since each member of $\mathcal{S}'$ can be formed by taking a member from $\mathcal{S}$ and add all radius values by a constant $\varepsilon/K$. Obviously, this procedure only makes the adversarial loss value larger and hence all the bounds still apply.

**Part II:** So far, we have managed to (partially) prove the proposed bound in the statement of the theorem in scenarios where $\min_k n_k \to \infty$ and thus we have $\widehat{P}_k \stackrel{a.s.}{=} P_k$ for all $k \in [K]$. At this stage of the proof we focus on replacing

$$\mathsf{QV}_k(h, \rho) = \sup_{Q \in \mathcal{B}_{\rho_k}^{\text{wass}}(P_k)} \mathbb{E}_Q\left[\ell\left(y, h\left(\boldsymbol{X}\right)\right)\right],$$

for any $k \in [K]$ and arbitrary $\rho \geq 0$, with its empirical version $\widehat{\mathsf{QV}}_k(h, \rho)$. Let us reiterate that we do not have any knowledge regarding $P_k$, and only client $k$ has access to its empirical version $\widehat{P}_k$ which is based on $n_k$ i.i.d. samples. Therefore, $\widehat{\mathsf{QV}}_k(h, \rho)$ is computable via the querying policy described in Section 4, while the true query value $\mathsf{QV}(h, \rho)$ is always unknown.

To this aim, similar to (Sinha et al., 2018) first let us define the following $\phi_\gamma(\boldsymbol{Z})$ function for $\gamma \geq 0$ and $\boldsymbol{Z} \in \mathcal{Z}$:

$$\phi_\gamma(\boldsymbol{Z}) \triangleq \sup_{\boldsymbol{Z}' \in \mathcal{Z}} \ell(\boldsymbol{Z}') - \gamma c(\boldsymbol{Z}', \boldsymbol{Z}), \tag{70}$$

where $c(\cdot, \cdot)$ is the original transportation cost and $\ell(\boldsymbol{Z})$ abbreviates $\ell(y, h(\boldsymbol{X}))$ where we have omitted $h$ for simplicity in notation. First, it can readily verified that if $\ell$ is bounded between 0 and 1, so does $\phi_\gamma$ for any $\gamma \geq 0$. Second, note that from Lemma C.1 we have the following duality formulation for $\mathsf{QV}_k$ and $\widehat{\mathsf{QV}}_k$ for any $\rho_k \geq 0$:

$$\mathsf{QV}_k(h, \rho_k) \triangleq \sup_{Q \in \mathcal{B}^{\mathrm{wass}}_{\rho_k}(P_k)} \mathbb{E}_Q[\ell(y, h(\boldsymbol{X}))] = \inf_{\gamma \geq 0} \{\gamma \rho_k + \mathbb{E}_{P_k}[\phi_\gamma(\boldsymbol{Z})]\},$$

$$\widehat{\mathsf{QV}}_k(h, \rho_k) \triangleq \sup_{Q \in \mathcal{B}^{\mathrm{wass}}_{\rho_k}(\widehat{P}_k)} \mathbb{E}_Q[\ell(y, h(\boldsymbol{X}))] = \inf_{\gamma \geq 0} \left\{\gamma \rho_k + \frac{1}{n_k} \sum_{i \in [n_k]} \phi_\gamma\left(\boldsymbol{Z}_i^{(k)}\right)\right\}. \tag{71}$$

In the following, first we show that $\phi_\gamma(\boldsymbol{Z})$, for any $\boldsymbol{Z} \in \mathcal{Z}$ is a Lipschitz function with respect to $\gamma$ where the Lipschitz constant only depends on the way the transportation cost $c$ is bounded, i.e., the inherent boundedness of $c$ or the compactness of $\mathcal{Z}$. Additionally, we show that the optimal $\gamma \geq 0$ in both minimization problems on the right-hand sides of equation 71 is bounded by a known constant. The latter result is deduced from the fact that all robustness radii $\rho_k$, $k \in [K]$ in the statement of theorem has a known margin from zero. Finally, we show the above-mentioned properties can guarantee that $\mathbb{E}_{\widehat{P}_k}[\phi_\gamma(\boldsymbol{Z})]$ uniformly converges to its true expected value $\mathbb{E}_{P_k}[\phi_\gamma(\boldsymbol{Z})]$ for all relevant value of $\gamma \geq 0$. Hence, the empirical and statistical query values are always within a controlled and asymptotically small deviation from each other regardless of the robustness radius value $\rho_k$.

In this regard, the following lemma shows that $\phi_\gamma(\boldsymbol{Z})$ for any $\boldsymbol{Z} \in \mathcal{Z}$ is a Lipschitz function with respect to $\gamma \geq 0$:

**Lemma C.3.** *There exists a constant $R \geq 0$ which only depends on transportation cost $c$ such that function $\phi_\gamma(\boldsymbol{Z})$ is $R$-Lipschitz with respect to $\gamma \geq 0$, for all $\boldsymbol{Z} \in \mathcal{Z}$.*

*Proof.* For any two distinct values $\gamma, \gamma' \geq 0$, let $\boldsymbol{Z}_\gamma^*$ and $\boldsymbol{Z}_{\gamma'}^*$ denote the optimal values for which the sup in equation 70 is attained. Similar to several previous arguments, attainability of the sup in this case is not necessary again, and thus this assumption is made for the sake of simplifying the proof.

Then, for any $\boldsymbol{Z} \in \mathcal{Z}$ we have:

$$\phi_\gamma(\boldsymbol{Z}) = \sup_{\boldsymbol{Z}' \in \mathcal{Z}} \ell(\boldsymbol{Z}') - \gamma c(\boldsymbol{Z}', \boldsymbol{Z}) \tag{72}$$

$$\geq \ell(\boldsymbol{Z}_{\gamma'}^*) - \gamma c(\boldsymbol{Z}_{\gamma'}^*, \boldsymbol{Z}),$$

$$\phi_{\gamma'}(\boldsymbol{Z}) = \ell(\boldsymbol{Z}_{\gamma'}^*) - \gamma' c(\boldsymbol{Z}_{\gamma'}^*, \boldsymbol{Z}),$$

which directly gives us the following bound:

$$\phi_\gamma(\boldsymbol{Z}) - \phi_{\gamma'}(\boldsymbol{Z}) \geq -(\gamma - \gamma') c(\boldsymbol{Z}_{\gamma'}^*, \boldsymbol{Z}). \tag{73}$$

Through a set of similar arguments and replacing $\gamma$ and $\gamma'$, the following complementary bound can be achieved as well:

$$\phi_\gamma(\boldsymbol{Z}) - \phi_{\gamma'}(\boldsymbol{Z}) \leq -(\gamma - \gamma') c(\boldsymbol{Z}_\gamma^*, \boldsymbol{Z}). \tag{74}$$

Therefore, the following inequality can be established according to the boundedness of $c$ (or alternatively, compactness of $\mathcal{Z}$):

$$
\begin{aligned}
|\phi_\gamma(\boldsymbol{Z}) - \phi_{\gamma'}(\boldsymbol{Z})| &\leq |\gamma - \gamma'| \max_{r \in \{\gamma, \gamma'\}} \{c(\boldsymbol{Z}_r^*, \boldsymbol{Z})\} \\
&\leq |\gamma - \gamma'| \sup_{\boldsymbol{Z}' \in \mathcal{Z}} c(\boldsymbol{Z}', \boldsymbol{Z}) \\
&\leq R|\gamma - \gamma'|,
\end{aligned}
\tag{75}
$$

which proves the Lipschitz-ness of $\phi_\gamma$ with respect to $\gamma$. $\qquad\square$

The following lemma shows that optimal values of $\gamma$ in the right-hand side minimization of equation 71 (or the infimum-achieving sequence in case the infimum is not attainable) is bounded by a known constant:

**Lemma C.4.** *In both minimization problems on the right-hand side of equation 71, the optimal $\gamma$ value denoted by $\gamma^*$ (if attained), or the tail of its sequence in case the* inf *is not attainable, satisfies* $0 \leq \gamma^* \leq \frac{1}{\rho_k}$.

*Proof.* Proof directly results from the fact that $\ell(\cdot)$ is bounded between $0$ and $1$. Therefore, looking at the dual optimization problem in equation 71, increasing $\gamma$ beyond $1/\rho_k$ results in $\gamma\rho_k > 1$ while the second term (i.e., the adversarial loss) is always lower-bounded by zero which makes the whole objective to become larger than 1. On the other hand, setting $\gamma = 0$ would (at worst) results in the objective to be 1. Therefore, the optimizer $\inf_{\gamma \geq 0}$ should not choose a $\gamma$ value that is larger than $1/\rho_k$. $\qquad\square$

At this point, we can state the main lemma in the second part of the proof, which theoretically shows that empirical query values, i.e., $\widehat{\mathsf{QV}}_k(h, \rho)$ for any fixed $h \in \mathcal{H}$ and uniformly all $\rho \geq 0$ converge to their true statistical expected values with a high probability.

**Lemma C.5** (Uniform Convergence of Empirical Queries). *For $k \in [K]$, assume the unknown sample distribution $P_k$ and let $\left\{ \boldsymbol{Z}_i^{(k)} = \left( \boldsymbol{X}_i^{(k)}, y_i^{(k)} \right) \right\}$ for $i \in [n_k]$ denote $n_k \in \mathbb{N}$ i.i.d. feature-label pairs drawn from $P_k$. The $k$th dataset is only known to client $k$. Then, for any fixed classifier $h$ and any $\delta > 0$, the following bound holds with probability at least $1 - \delta/(K+2)$:*

$$
\sup_{\rho \geq \varepsilon/K} \left| \widehat{\mathsf{QV}}_k(h, \rho) - \mathsf{QV}_k(h, \rho) \right| \leq \mathcal{O}\left( \sqrt{\frac{\log\left(\frac{(K+2)n_k}{\varepsilon\delta}\right)}{n_k}} \right).
\tag{76}
$$

*Proof.* Using the dual formulation of Lemma C.1, we can rewrite the main objective of the theorem as follows:

$$
\begin{aligned}
&\sup_{\rho \geq \varepsilon/K} \left| \widehat{\mathsf{QV}}_k(h, \rho) - \mathsf{QV}_k(h, \rho) \right| \\
&= \sup_{\rho_k \geq \varepsilon/K} \left\{ \inf_{\widehat{\gamma} \geq 0} \left[ \widehat{\gamma}\rho_k + \mathbb{E}_{\widehat{P}_k}[\phi_{\widehat{\gamma}}(\boldsymbol{Z})] \right] - \inf_{\gamma \geq 0} \left[ \gamma\rho_k + \mathbb{E}_{P_k}[\phi_\gamma(\boldsymbol{Z})] \right] \right\}.
\end{aligned}
\tag{77}
$$

Again, for the sake of simplicity in the proof let us assume both optimal values $\gamma^*$ and $\widehat{\gamma}^*$ in the minimization problems on the right-hand side of equation 77 are attainable. It should be noted that this assumption is not necessary and can be relaxed by adding more mathematical work. Then, from Lemma C.4 we already know

$$
0 \leq \gamma^*, \widehat{\gamma}^* \leq \frac{1}{\rho_k} \leq \frac{K}{\varepsilon}.
$$

Let us partition the feasible search set of $\gamma, \widehat{\gamma} \geq 0$, i.e., $[0, K/\varepsilon]$ into $L \triangleq \lceil \frac{K^2 R}{\varepsilon\Delta} \rceil$ equal intervals, where $\Delta > 0$ is a small constant which should to be determined later in the proof. For each interval, let us choose a representative (for example, the value at the beginning of the interval) denoted by

$\gamma_i$ with $i = 1, \ldots, L$. Then, based on Lemma C.3, for any $\rho_k \in [\varepsilon/K, 2K\varepsilon]$, any corresponding $\gamma \in [0, 1/\rho_k]$ and all $\boldsymbol{Z} \in \mathcal{Z}$ we have

$$|\phi_\gamma(\boldsymbol{Z}) - \phi_{\gamma_{i*}}(\boldsymbol{Z})| \leq R|\gamma - \gamma_{i*}| \leq R \cdot \frac{K}{\varepsilon} \cdot \frac{\varepsilon\Delta}{K^2 R} = \frac{\Delta}{K}, \tag{78}$$

where $i^* = \arg\min_{i \in [L]} |\gamma - \gamma_i|$. On the other hand, from the maximization problem that defines $\widehat{U}^*(\varepsilon)$ in equation 12, we have that

$$\rho_k \leq \mathcal{O}\left(K\varepsilon + \sqrt{K \log\left(\frac{K+2}{\delta}\right)}\right), \ \forall k \in [K], \tag{79}$$

where we have omitted constants for the sake of readability. Therefore, the above discussions directly lead to the following bound in the statistical sense (i.e., with respect to $P_k$):

$$\left| \inf_{\gamma \geq 0} \ [\gamma\rho_k + \mathbb{E}_{P_k}[\phi_\gamma(\boldsymbol{Z})]] - \min_{i \in [L]} \ [\gamma_i\rho_k + \mathbb{E}_{P_k}[\phi_{\gamma_i}(\boldsymbol{Z})]] \right|$$

$$\leq \ \min_{i \in [L]} |\gamma^* - \gamma_i| (R + \sup \ \rho_k)$$

$$\leq \ \frac{K}{\varepsilon} \cdot \frac{\varepsilon\Delta}{K^2 R} \cdot \mathcal{O}\left(R + K\varepsilon + \sqrt{K \log\left(\frac{(K+2)}{\delta}\right)}\right)$$

$$\leq \ \Delta \cdot \mathcal{O}\left(\frac{1}{K} + \frac{\varepsilon}{R} + \frac{1}{R}\sqrt{\frac{\log\left(\frac{K+2}{\delta}\right)}{K}}\right). \tag{80}$$

Through a similar procedure, the following bound also holds for the empirical case (i.e., $\widehat{P}_k$), but this time *almost surely*:

$$\left| \inf_{\widehat{\gamma} \geq 0} \ \left[\widehat{\gamma}\rho_k + \mathbb{E}_{\widehat{P}_k}[\phi_{\widehat{\gamma}}(\boldsymbol{Z})]\right] - \min_{i \in [L]} \ \left[\gamma_i\rho_k + \mathbb{E}_{\widehat{P}_k}[\phi_{\gamma_i}(\boldsymbol{Z})]\right] \right|$$

$$\overset{a.s.}{\leq} \ \Delta \cdot \mathcal{O}\left(\frac{1}{K} + \frac{\varepsilon}{R} + \frac{1}{R}\sqrt{\frac{\log\left(\frac{K+2}{\delta}\right)}{K}}\right). \tag{81}$$

Therefore, according to the fact that $\widehat{P}_k$ is an empirical estimate of $P_k$ based on $n_k$ i.i.d. samples , we have:

$$\sup_{\rho \geq \varepsilon/K} \ \left|\widehat{\mathsf{QV}}_k(h, \rho) - \mathsf{QV}_k(h, \rho)\right| \overset{a.s.}{\leq} \ \Delta \cdot \mathcal{O}\left(\frac{1}{K} + \frac{\varepsilon}{R} + \frac{1}{R}\sqrt{\frac{\log\left(\frac{K+2}{\delta}\right)}{K}}\right) +$$

$$\max_{i \in [L]}\left|\mathbb{E}_{\widehat{P}_k}[\phi_{\gamma_i}(\boldsymbol{Z})] - \mathbb{E}_{P_k}[\phi_{\gamma_i}(\boldsymbol{Z})]\right|. \tag{82}$$

Since $\phi_{\gamma_i}$ for each $i \in [L]$ is a non-negative and 1-bounded (adversarial) loss function, simply applying McDiarmid's inequality and the union bound over all $L$ values of $\gamma_i$s would give us the following bound which holds with probability at least $1 - \delta/(K+2)$ for any $\delta > 0$:

$$\max_{i \in [L]}\left|\mathbb{E}_{\widehat{P}_k}[\phi_{\gamma_i}(\boldsymbol{Z})] - \mathbb{E}_{P_k}[\phi_{\gamma_i}(\boldsymbol{Z})]\right| \leq \sqrt{\frac{\log\left[\frac{L(K+2)}{\delta}\right]}{2n_k}}, \tag{83}$$

which gives us the following bound for $\sup_{\rho \geq \varepsilon/K} \left|\widehat{\mathsf{QV}}_k(h, \rho) - \mathsf{QV}_k(h, \rho)\right|$ with the same high probability:

$$\sup_{\rho \geq \varepsilon/K} \ \left|\widehat{\mathsf{QV}}_k(h, \rho) - \mathsf{QV}_k(h, \rho)\right| \leq \mathcal{O}\left(\inf_{\Delta > 0} \left\{\Delta\zeta(K, \varepsilon, \delta) + \sqrt{\frac{\log\left[\frac{RK^2(K+2)}{\varepsilon\delta\Delta}\right]}{n_k}}\right\}\right), \tag{84}$$

where function $\zeta(\cdot)$ is defined as

$$\zeta(K, \varepsilon, \delta) \triangleq \frac{1}{K} + \frac{\varepsilon}{R} + \frac{1}{R}\sqrt{\frac{\log\left(\frac{K+2}{\delta}\right)}{K}}.$$

It should be noted that $R$ is a constant that does not depend on other parameters. Also, we have minimized over $\Delta > 0$ since the bound holds irrespective of $\Delta$. Exact solution of the minimization problem in equation 84 is not needed, since choosing $\Delta = \mathcal{O}\left(K^{-1}n_k^{-1/2}\right)$ gives us the following bound:

$$\sup_{\rho \geq \varepsilon/K} \left|\widehat{\mathsf{QV}}_k(h, \rho) - \mathsf{QV}_k(h, \rho)\right| \leq \mathcal{O}\left(\sqrt{\frac{\log\left(\frac{(K+2)n_k}{\varepsilon\delta}\right)}{n_k}}\right), \tag{85}$$

and completes the proof. $\qquad\square$

By using the uniform convergence result from Lemma C.5 and applying it to all $K$ clients simultaneously, we see that (via a union bound argument) with probability at least $1 - K\delta/(K+2)$ the empirical and statistical queries $\widehat{\mathsf{QV}}_k(h, \rho_k)$ and $\mathsf{QV}_k(h, \rho_k)$ are asymptotically close for all $k \in [K]$, and uniformly for all robustness radii

$$\rho_k \in \left[\frac{\varepsilon}{K}, \mathcal{O}\left(K\varepsilon + \sqrt{\frac{\log((K+2)/\delta)}{K}}\right)\right]$$

which are considered for the maximization problem of equation 12. Finally, using another union bound argument to incorporate the bounds from equation 64 and equation 67 in addition to the previous $K$ events, we can say that with probability at least $1 - \delta$ the following bound holds:

$$\sup_{\mu' \in \mathcal{G}_\varepsilon(\mu)} \mathbb{E}_{P \sim \mu'}\left[\mathbb{E}_P\left[\ell(y, h(\boldsymbol{X}))\right]\right] \tag{86}$$

$$\leq \sup_{\underline{\rho} \in \mathcal{S}'} \frac{1}{K} \sum_{k \in [K]} \sup_{Q \in \mathcal{B}_{\rho_k}^{\mathrm{wass}}(P_k)} \mathbb{E}_Q\left[\ell(y, h(\boldsymbol{X}))\right] + c_2\sqrt{\frac{\log\left(\frac{(K+2)n_k}{\varepsilon\delta}\right)}{n_k}} + \sqrt{\frac{\log\left(\frac{K+2}{\delta}\right)}{2K}}$$

$$= \sup_{\underline{\rho} \in \mathcal{S}'} \frac{1}{K} \sum_{k \in [K]} \widehat{\mathsf{QV}}_k(h, \rho_k) + c_2\sqrt{\frac{\log\left(\frac{(K+2)n_k}{\varepsilon\delta}\right)}{n_k}} + \sqrt{\frac{\log\left(\frac{K+2}{\delta}\right)}{2K}}, \tag{87}$$

where $c_2$ is a constant that only depends on either the transportation cost $c$ or the compactness of sample space $\mathcal{Z}$. This completes the proof. $\qquad\square$

# D  AUXILIARY PROOFS: REMARKS, LEMMAS, ETC.

*Proof of Remark 6.3.* The query functions $\widehat{\mathsf{QV}}_k(h, \rho)$, for any fixed $h \in \mathcal{H}$, are non-decreasing with respect to $\rho \geq 0$. Therefore, the following function:

$$\zeta(\rho_1, \ldots, \rho_K) \triangleq \frac{1}{K} \sum_{k \in [K]} \widehat{\mathsf{QV}}_k(h, \rho_k) \tag{88}$$

is a summation of $K$ non-decreasing functions, where each function depends only on one of the $\rho_k$ values. This function, $\zeta(\rho_1, \ldots, \rho_K)$, is a *quasi-concave* function. Although quasi-concave functions are not generally concave, they do possess concave superlevel sets. Specifically, for each $t \in \mathbb{R}$, the following sets are convex in $\mathbb{R}^K$:

$$\mathcal{S}_t \triangleq \left\{(\rho_1, \ldots, \rho_K) \in \mathbb{R}^K \,\middle|\, \frac{1}{K} \sum_{k \in [K]} \widehat{\mathsf{QV}}_k(h, \rho_k) \geq t\right\}. \tag{89}$$

---

**Algorithm 1:** Server-side Bisection Algorithm

---

**Parameters:** $K, \varepsilon, \delta$

**Input**      : $h, \Delta > 0$, $\text{poly}\left(K, \log \frac{1}{\Delta}\right)$ query budget for $\widehat{\mathsf{QV}}_k$, $\forall k \in [K]$

**Initialize:**
> $a \longleftarrow \min \ell(\cdot) \text{ or } 0$
> $b \longleftarrow \max \ell(\cdot) \text{ or } 1$

**while** $b - a > \Delta$ **do**
> $t \longleftarrow (a + b)/2$
>
> Solve convex feasibility problem :
>
>   Find $\rho_1, \ldots \rho_K \geq \varepsilon/K$
>
>   subject to  $\dfrac{1}{K} \sum_{k \in [K]} \rho_k \leq \varepsilon \left(1 + \dfrac{1}{K}\right) + c_1 \sqrt{\dfrac{\log\left((K+2)/\delta\right)}{K}}$
>
>                           $\dfrac{1}{K} \sum_{k \in [K]} \widehat{\mathsf{QV}}_k (h, \rho_k) \geq t$
>
> **if** *Is feasible* **then**
> > $a \longleftarrow t$
>
> **else**
> > $b \longleftarrow t$

**Result:** Upper-bound $b$

---

As a result, the original optimization problem in equation 12 can be decomposed into a sequence of convex optimization problems. Each sub-problem is essentially a feasibility problem that checks whether the set $\mathcal{S}_t$ is feasible for a given $t \in \mathbb{R}$. Given that the original objective function is bounded between $0$ and $1$, a binary search can be employed to iteratively approximate the maximum attainable value of the objective within any desired error margin $\Delta > 0$. This process is detailed in Algorithm 1, which implements a *bisection* algorithm.

It is important to note that each feasibility check sub-problem in Algorithm 1 is a convex problem, and therefore can be solved in polynomial time with polynomial evaluations of the constraint functions, i.e., the $\widehat{\mathsf{QV}}_k$ functions. Consequently, both the computational complexity and the required query budget remain polynomial. In particular, for a given $\Delta > 0$, the maximum of the objective in equation 88 can be approximated with an error of at most $\Delta$, requiring at most $\log \frac{1}{\Delta}$ iterations (feasibility checks) in Algorithm 1. $\qquad\square$

# E    AUXILIARY ILLUSTRATIONS AND EXPERIMENTAL RESULTS

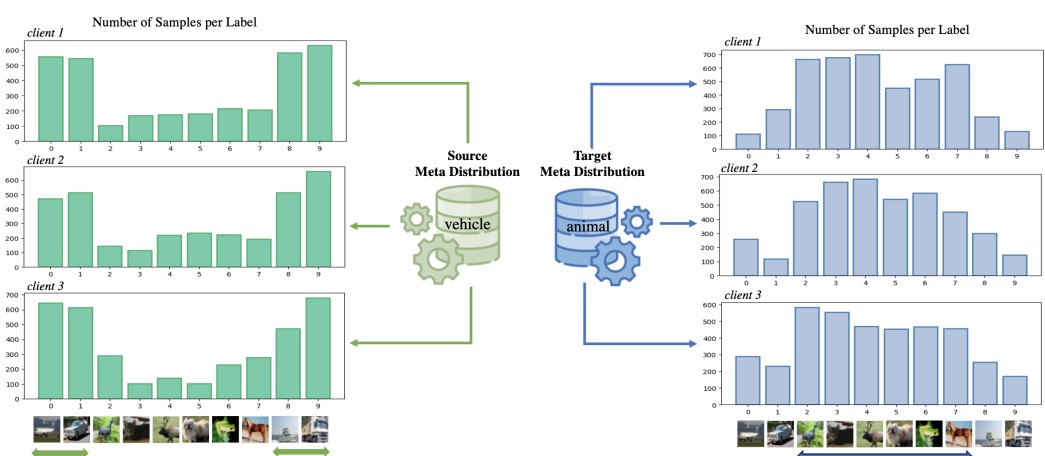

Figure 4:   A graphical illustration of meta-distributional shifts between two different networks (or societies) of clients/users. Heterogeneous user data distributions can be considered as i.i.d. samples from a meta-distribution. In this example, clients from one meta-distribution primarily have vehicle images, while those from another meta-distribution mostly have animal images.

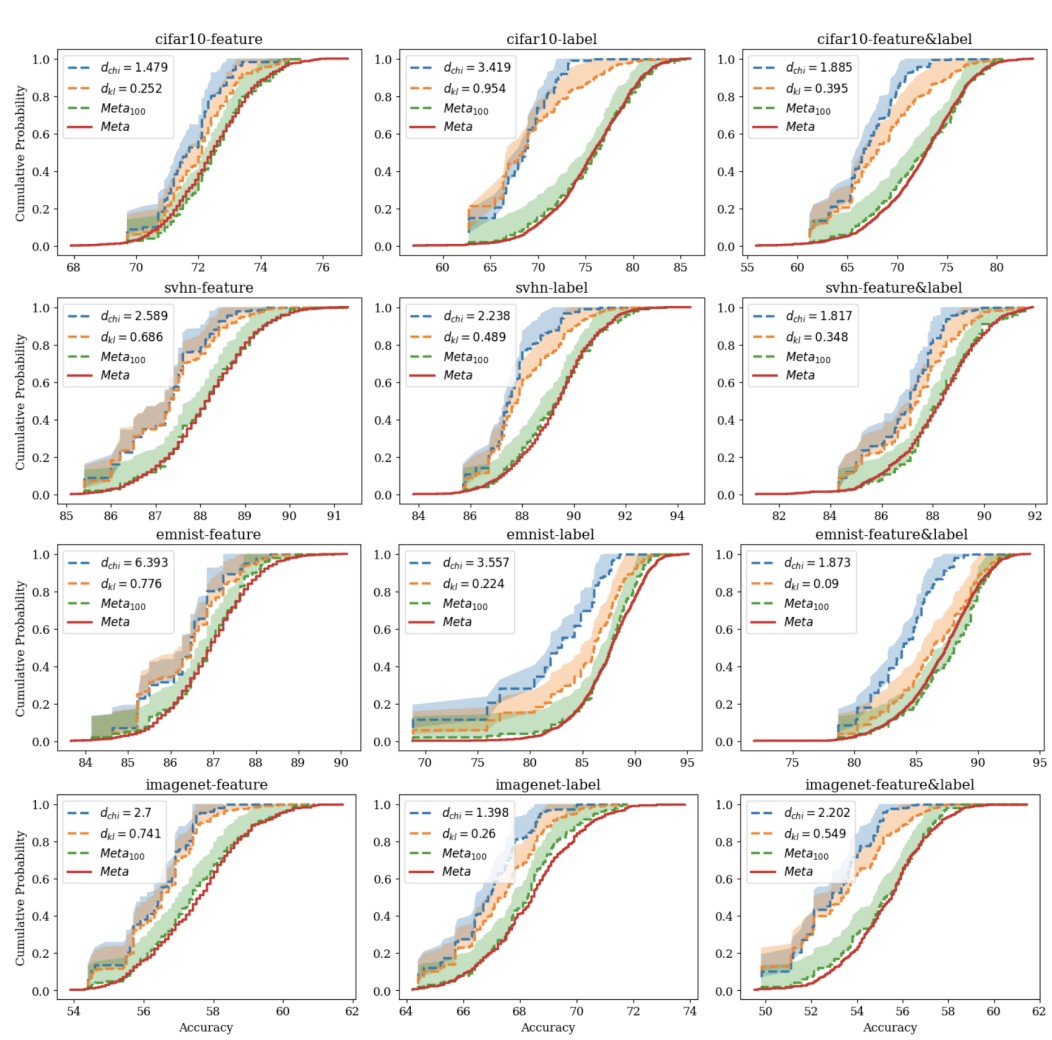

Figure 5: Extension of previous experiments to a broader range of adversarial budgets for $f$-divergence attacks. DKW-based certificates for unseen clients in our four examined datasets. *Meta* refers to the main population with 500 nodes.

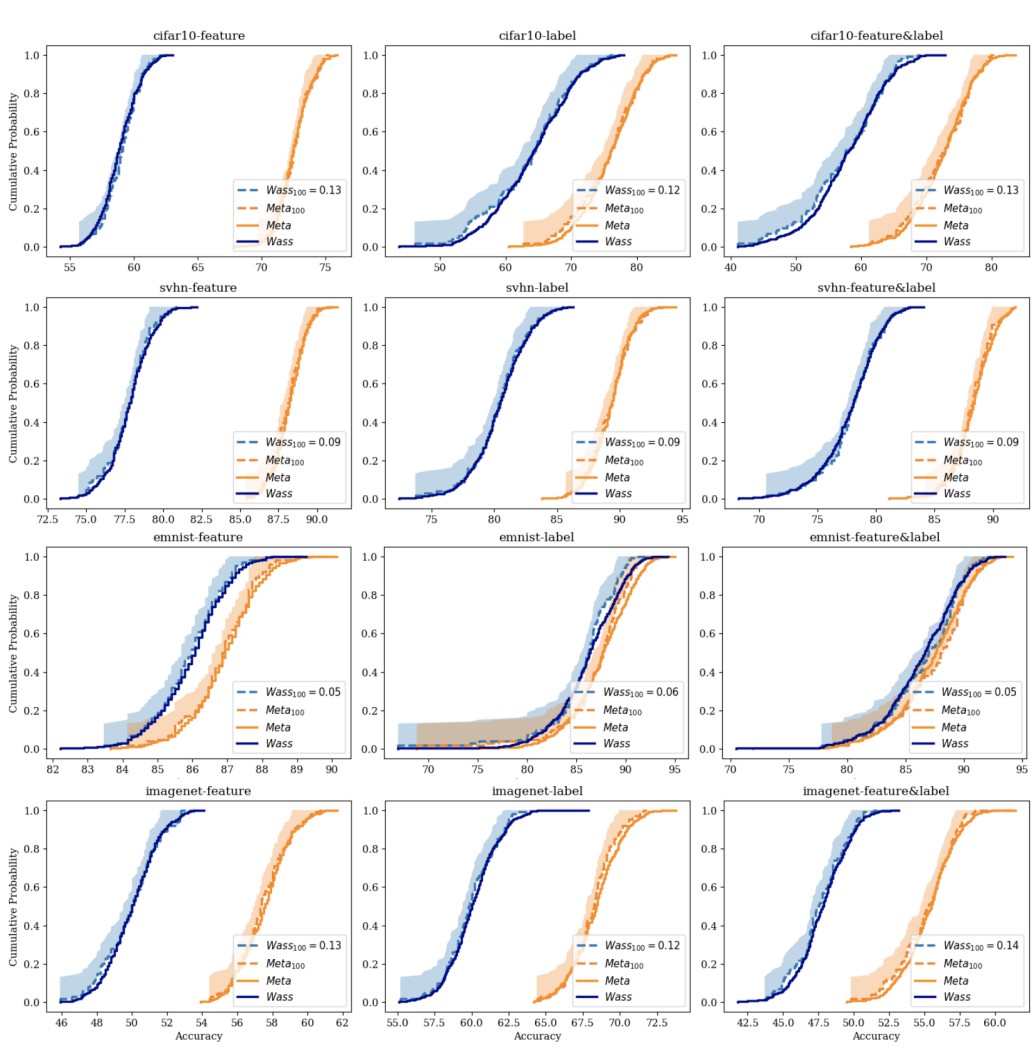

Figure 6: Wasserstein distance-based certificates for unseen clients in our four examined datasets. *Meta* and *Wass* refer to the main population with 500 nodes. Dotted curves are based on 100 networks within the population.

