# OpenReview forum: "Robust Model Evaluation over Large-scale Federated Networks"
_ICLR.cc/2025/Conference — Submitted to ICLR 2025_

### Official Review · Reviewer_oDR9 · 2024-10-27

**Soundness:** 2
**Presentation:** 1
**Contribution:** 2
**Rating:** 3
**Confidence:** 2

**Summary:**

This paper considers the problem of certifying a model’s performance over a federated clients drawn from an unseen meta distribution that is close to a set of federated clients drawn from a seen meta distribution.
The theoretical results for certification are given for Wasserstein distance and f-divergence respectively with the help of Glivenko-Cantelli theorem and the Dvoretzky-Kiefer-Wolfowitz (DKW) inequality. Such certification can be computed within polynomial time.

**Strengths:**

- Having certificates of model’s performance can be important in practice.

**Weaknesses:**

The paper is presented in a way which is very difficult to understand:

- Important definitions and assumptions are deferred to the appendix.
- Theoretical results are given without much explanation. I don’t know why B^hat is subject in the current form and it is hard to tell if the theorems in the paper makes sense or not.
- Why for f-divergence we use QVs with rho=0?
- The name of Definition 5.1 is “meta-distributional f-divergence Ball”,  but the equation (4) additionally assume the Lambda-bounded density ratio which means you don’t actually prove for the entire f-divergence ball. Despite the descriptions above the definition talks about what does Lambda-bounded density ratio do, it is not clear to me is this really necessary or is it just another assumption that makes your proof easier?
- The “Lambda-bounded density ratio” assumption is listed in the appendix without any references. Is this new or already exists in the literature. You need to either cite previous works or justify this assumption is realistic.
- What exactly are c_1 and c_2 in Eq (5) since we need to approximate B^\hat?
- For f-divergence, distributions sharing same support should be explicitly listed as an assumption.
- It is claimed that the term in line 220 decreases with increasing K and \min n_k. However, it does not specify how fast \min n_k grows. If \log K outgrows  \min n_k then the term does not decrease.
- Line 235 “Additionally, any inherent “wellness” or “robustness” of the target model h is directly reflected through a low value for Bb∗(ε)” There is no guarantee that B(\epsilon) and QK is small.
- Certificate of Privacy: I doubt the significance of the claims here. Data recovery attacks are targeting the training where gradients are exchanged in each iteration whereas the task here is to estimate the loss here. Not really fair comparison.

Therefore I don’t think this paper in the current form is ready for publication.

**Questions:**

See above

---

> ### Author Response · Authors · 2024-11-16
> **Response to Reviewer oDR9**
>
> We thank the reviewer for their time and thoughtful feedback. Below, we address each of the reviewer's comments in detail:
>
> - **“Important definitions and assumptions are deferred to the appendix.”** Due to space limitations, we had to move some foundational definitions to the appendix to make room for more technical content. We will adjust this in the revised version. Please also refer to our response to Reviewer 9X7d.
>
> - **“I don’t know why B^hat is subject in the current form.”** Consider the task of estimating the mean of a distribution from a sample set. Typically, one would use the sample (empirical) mean. However, the question here is: what would be the worst-case estimate of the mean if samples come from a different but nearby distribution, in terms of f-divergence? Intuitively, one could reweight the empirical mean by giving larger samples more weight, with weights constrained to remain close to 1. This illustrates the idea behind $\hat{B}$; we mathematically prove that this reweighted estimator is sufficient (achieving a vanishing generalization gap).
>
> - **“Why for f-divergence we use QVs with ρ=0?”** This is one of our interesting findings. To achieve robust model evaluation under a bounded $f$-divergence shift in the meta-distribution, clients do *not* need to compute a private adversarial loss (hence, $\rho = 0$, implying the ordinary loss is sufficient). The adversarial aggregation occurs solely on the server side. In contrast, under the Wasserstein distance, clients must compute their adversarial losses according to optimal adversarial budgets, which the server determines. The server finds these optimal budgets in polynomial time, requests adversarial losses from clients, and then aggregates them.
>
> - **“The name of Definition 5.1 is ‘meta-distributional f-divergence Ball,’ but equation (4) additionally assumes a Λ-bounded density ratio.”** When working with $f$-divergences, assuming bounded density ratios can simplify the analysis. While this assumption can be removed, doing so would introduce additional technical complexity. Furthermore, the bounds become dependent on the choice of the function $f$. For example, without the $\Lambda$-bounded assumption, the bound for the particular case of KL-divergence changes from $\Lambda \cdot K^{-1/2}$ to $\sqrt{\log K / K}$. However, it may slightly change for example if we use chi-square function. We will add references to similar assumptions used in prior work to improve clarity.
>
> - **“What exactly are $c_1$ and $c_2$ in Eq (5) since we need to approximate $B^{\hat{}}$?”** Constants $c_1$ and $c_2$ result from multiple applications of McDiarmid's inequality and the union bound. Using basic mathematics, both can be shown to be bounded by 32. However, in practice, smaller values (approximately 2) suffice to keep the bounds tight across our experiments. Thus, finding the optimal values for $c_1$ and $c_2$ is left as future work. We will add explanations regarding this in the revised version.
>
> - **“For f-divergence, distributions sharing the same support should be explicitly listed as an assumption.”** In Definition A.1, we state that $\mu'$ must be absolutely continuous with respect to $\mu$, which basically means the same thing is reviewer suggested.
>
> - **“It does not specify how fast $\min n_k$ grows. If $\log K$ outgrows $\min n_k$, then the term does not decrease.”** We have already emphasized in line 356 that $(\min n_k) / \log K$ needs to grow. We will also clarify other statements that may be ambiguous on this point.

---

> > ### Author Response · Authors · 2024-11-16
> >
> > - **Line 235: Additionally,  any inherent wellness or robustness of the target model $h$ is directly reflected through a low value for $B_{b}^{*}(\epsilon)$ - There is no guarantee that $B(\epsilon)$ and $QK$ are small.**  Please note that our bounds are asymptotically minimax optimal. This implies that if these values are large for a given $\epsilon$ (as assumed by the reviewer), then for sufficiently large $K$ and $n_k$, there exists a malign meta-distribution $\mu'$ that can actually attain the bound, or get extremely close to it.
> >
> > - **Certificate of Privacy: I doubt the significance of the claims here. Data recovery attacks target training where gradients are exchanged in each iteration, whereas the task here is to estimate the loss.** The reviewer is correct. We did not intend to draw an unfair comparison. The purpose of this remark is to emphasize that our algorithm is resistant to data leakage based on known attacks. Why is this noteworthy? In traditional methods, achieving an adversarially robust loss often requires pooling all datasets at the server side to perform global adversarial perturbations. Our approach shows that applying local, private perturbations and then combining them in polynomial time yields comparable results. To our knowledge, this work is the first to demonstrate this approach for federated model evaluation (also please refer to our response to other reviewers).
> >
> > We hope these responses address the reviewer’s concerns, and we respectfully request a reconsideration of the score, as we believe the current score does not fully reflect the contributions and effort invested in this work.

---

> > > ### Author Response · Authors · 2024-11-22
> > >
> > > Dear Reviewer oDR9,
> > >
> > > Since we are approaching the end of the discussion period, we appreciate it if you could give us your feedback after reading the rebuttal.
> > >
> > > Thanks in advance,
> > >
> > > Authors

---

> > > ### Comment · Reviewer_oDR9 · 2024-11-26
> > > **Reply**
> > >
> > > I acknowledge that I have read authors' response and appreciate authors' effort in addressing the concerns. The paper in the current form still looks premature to me. I have decided to keep my score.

---

### Official Review · Reviewer_9X7d · 2024-11-02

**Soundness:** 3
**Presentation:** 4
**Contribution:** 3
**Rating:** 6
**Confidence:** 4

**Summary:**

This paper provides certified guarantees for the performance of a model, which is trained on a federated learning network, on an unseen network/clients. The data distribution of the clients in the unseen network follow a meta distribution with bounded deviation from the meta distribution of the training network. The deviation is measured in terms of f-divergence and Wasserstein distance. The upper bounds provided for the performance of the model are in terms of empirical average loss (for both the deviation measures above) and CDF of the loss (for f-divergence only). As the main contribution of the work, the generalization gap in the derived bounds vanishes as number of clients and their dataset sizes grow.

**Strengths:**

* Despite being a theoretical paper, it is well-written and it is easy to read.
* The work is theoretically rigorous with helpful and easy-to-read explanations about each part.
* The experimental results clearly support the theoretical derivations in the work.

**Weaknesses:**

There are some weaknesses about the work:
* The existing results in the literature that are closely related to this work are not discussed enough.

* The authors claim (line 227) that the main advantage of their work is that the generalization gaps in their derived bounds do not have a non-vanishing term, unlike many existing bounds. However, they have not compared their results with at least the best result that exists in the literature. This could be discussed in a separate paragraph after each of the Theorems 5.2, 5.5 and 6.2 (or even deferred to the appendix). Such a discussion would provide a more clear and convincing view of the main contribution and novelty of this work. This could be extended to the experimental results too.

* There are some experimental results that could improve the draft. I have asked about them in the following questions.

**Questions:**

Other than the weaknesses mentioned above, which are recommended to be addressed, I have some questions as follows:

* In all the three main theorems, the clients are queried by the server about their "expected" non-robust or robust risk values. What is the main benefit of knowing $n_k$ for the server? From the theorems, we see that $n_k$ appears in either the server's queries (in theorem 5.5) or in the generalization gaps (in theorems 5.2 and 6.2) (and not both). Can we eliminate the $n_k$ from the server's queries and replace them with a lower bound for them? This way, the server would just need a lower bound to do its queries without even knowing the clients' dataset sizes, hence providing some extra privacy. Remember that in some FL settings, the aggregation weights for clients are not based on their dataset sizes, but based on their empirical average risk, e.g., in some FL algorithms addressing performance fairness [1]. In such algorithms, the server does not know the clients' dataset sizes.

[1] Tian Li, Ahmad Beirami, Maziar Sanjabi, and Virginia Smith. Tilted empirical risk minimization. In International Conference on Learning Representations, 2020.


*  Unlike in theorems 5.2 and 5.5, where the query budget is one for each client, in theorem 6.2, the server queries clients about their robust risk values with "polynomially many" values of $\rho$ (robustness radii). What is the value of query budget considered for this case in your experiments? (e.g., in Fig. 3 right). Also, I am wondering how this would affect the bound computed in your experiments? An ablation study on the query budget, when ranging from 1 (similar to theorems 5.2 and 5.5) to larger values would reveal its behaviour when the "computational complexity/communication budget" changes.


* What exactly is the difference between Fig. 1 and Fig. 5? From what I see they are the same. How exactly the "adversarial budget" changes between the two?


Minor comments:
* line 063: "both" should be dropped?
* line 093: there are two "the"s

---

> ### Author Response · Authors · 2024-11-16
> **Response to Reviewer 9X7d**
>
> We would like to thank the reviewer for their careful review and thoughtful feedback. Below, we provide point-by-point responses to each of the reviewer’s concerns:
>
> - **“Literature review and comparison with other theoretical works is not enough.”**
>   In the revised version, we will address this issue. In our initial submission, we had to shorten the introductory sections to accommodate the technical results. In the revision, we will move some of these technical details to the appendix, allowing us to expand the literature review and provide additional explanations and clarifications in the earlier sections.
>
> - **“What is the main benefit of knowing $n_k$ for the server?”**
>   It is not essential for computing the empirical bounds. The values of $n_k$ are only needed if one aims to bound the generalization gap. We can therefore remove the necessity of knowing $n_k$ from our requirements. Thank you for this suggestion!
>
> - **“What is the value of the query budget considered for this case in your experiments?”**
>   In our experiments, we implemented a simple heuristic search algorithm to find robustness radii (specifically, their dual counterparts, the $\gamma$ values) rather than theoretically-guaranteed oracle-based optimization methods. We did not elaborate further on this approach, as we observed that the bounds became tight quickly even with this heuristic search. We will add more detail about this in the revised version.
>
> - **“What exactly is the difference between Fig. 1 and Fig. 5?”**
>   In Fig. 1, we present results for the CIFAR and ImageNet datasets, while Fig. 5 includes additional results from the SVHN and EMNIST datasets. Displaying all four datasets together makes the results more comparable. Both figures correspond to $f$-divergences with different choices of the function $f$ and various values of $\epsilon$, which represents our adversarial budget.
>
> - We also appreciate the reviewer pointing out typographical errors, which we will correct in the revised version.
>
> We hope these responses address the reviewer’s concerns and would be grateful if they could raise their score.

---

> > ### Author Response · Authors · 2024-11-22
> >
> > Dear Reviewer 9X7d,
> >
> > Since we are approaching the end of the discussion period, we appreciate it if you could give us your feedback after reading the rebuttal.
> >
> > Thanks in advance,
> > Authors

---

> > > ### Comment · Reviewer_9X7d · 2024-11-23
> > > **Thank you for the response**
> > >
> > > Thank you for the answers. After reading the answers, I have decided to keep my score.

---

### Official Review · Reviewer_Vduy · 2024-11-03

**Soundness:** 2
**Presentation:** 2
**Contribution:** 2
**Rating:** 5
**Confidence:** 2

**Summary:**

This paper studies the problem of model evaluation on unseen clients. The assumption is that each client is associated with a data distribution, and these distributions are from a meta distribution. The evaluation goes as follows: the server requests the client to evaluate a model on its private dataset, and the client returns an adversarial risk. The goal is to provide an upper bound for the risk of the model. Theoretical bounds are provided with assumption that the unseen clients have close data distribution, characterized by f-divergence or Wasserstein distance. Experiments are conducted to support the theoretical claims.

**Strengths:**

The proof involves multiple advanced results in robust estimation and looks solid.

**Weaknesses:**

- It seems that this paper is purely theoretical and it is not clear if the work has real-world applications. For instance, the assumption is that the data distribution for the unseen clients is close in f-divergence or Wasserstein distance. What does this mean for heterogeneous data distribution? Does this assumption work with non-IID data distribution generation in previous research like [1]? Also, the client needs to return a query value with complicated forms; is it feasible to compute? It is a supremum of expectations.

- It is hard to parse the experiment results. There is no comparison with other methods, making it difficult to evaluate the novelty of this work. Multiple CDF curves are provided, but it is not clear the meaning of the curves. Is the goal to show the correctness of Theorem 5.5 and Theorem 6.2? But these are asymptotic bounds, and it is not clear to me why you can use experiments to show the correctness.


[1] Karimireddy, Sai Praneeth, et al. "Scaffold: Stochastic controlled averaging for federated learning." International conference on machine learning. PMLR, 2020.

**Questions:**

- Why is this work related to Federated learning? As far as I can see, the main result is saying if we evaluate a model on some clients, and another client has data distribution that is close enough to those clients, then we can have some bounds on the model risk on this client. The model itself can be arbitrary and does not need to be trained with some FL algorithm. Can the authors elaborate a bit more on the connection with federated learning?
- The paper also mentions "large-scale" in the title. Is it indeed relevant? The evaluation algorithm does not seem to depend on model structure or can be accelerated for large neural networks.
- Text in Figure 3 is too small to read.

---

> ### Author Response · Authors · 2024-11-16
> **Response to Reviewer Vduy**
>
> We thank the reviewer for their time and thoughtful feedback. Below, we address each of the reviewer's comments in detail:
>
> 1. **"The assumption is that the data distribution for the unseen clients is close in f-divergence or Wasserstein distance.":** We believe there may be a misunderstanding here. We do **not** assume that client distributions are close in any sense; indeed, two clients may have very distant data distributions (in f-divergence or Wasserstein distance). In this respect, our setting aligns with that of Scaffold [1], as the reviewer mentions. In our work, we assume that distributions for the source network are sampled from a meta-distribution $\mu$, while those for the target network come from a different meta-distribution $\mu'$. We assume $\mu$ and $\mu'$ are close, but do not impose any closeness between individual distributions sampled from either of them. Otherwise, providing guarantees on the target network based only on source observations is mathematically infeasible. We will add further clarification to the revised version.
>
> 2. **"The client needs to return a query value with complicated forms; is it feasible to compute? It is a supremum of expectations.":** Yes, fortunately, DRO problems of this type are theoretically and practically solvable in polynomial time. The seminal work of Sinha et al. (2017) demonstrates that, in the empirical regime, such problems have a dual form that is concave and solvable in polynomial time. Their work has been extensively used in various robust optimization applications since then. We will emphasize this result more explicitly in the revised version.
>
> 3. **"It is hard to parse the experiment results. There is no comparison with other methods.":** We have two responses: (i) Our CDF bounds are novel, and to the best of our knowledge, there are no previous works providing bounds on the loss CDF. On the other hand, our average loss bounds are already shown to be tight, so we did not include a comparison in that regard. The experiments aim to demonstrate the tightness of our bounds rather than improvements over existing methods. (ii) Our work is mostly theoretical, and our primary goal in the experimental section is to assure readers of the soundness and computability of our bounds.
>
> 4. **"Is the goal to show the correctness of Theorem 5.5 and Theorem 6.2? But these are asymptotic bounds.":** All our theorems, except for Corollary A.4, are **non-asymptotic**, meaning they hold for all values of $K$ and $n_k$.
>
> **Questions:**
>
> - **"Why is this work related to federated learning? As far as I can see, the main result is saying if we evaluate a model on some clients, and another client has a data distribution that is close enough to those clients, then we can have some bounds on the model risk on this client.":** As mentioned above, we do *not* assume a single target user with a distribution close to the source distributions; rather, the closeness applies at the level of meta-distributions, not individual user distributions. Federated model evaluation, much like federated model training, is an active research area focused on providing average or tail bounds on model performance using loss values from a limited number of clients without directly accessing their data.
>
>     Attaining the "robust" loss or CDF tail typically requires centralizing heterogeneous datasets on a server to perturb empirical data and estimate adversarial loss, which can compromise privacy. In contrast, our approach shows that robust average loss/loss CDF values can be attained without pooling datasets. Each user can perform a private adversarial risk assessment (based on an adversarial budget set by the server), and the server can then combine these results in a privacy-preserving manner.
>
>     Additionally, the server’s interactions with clients follow policies governed by QV (query value) functions. For shifts in $f$-divergence, the server aggregates the values of QV in an adversarial manner. In the case of Wasserstein distance, the server identifies the optimal adversarial budget for each client in polynomial time, then aggregates the results accordingly.
>
> - **"The paper also mentions 'large-scale' in the title. Is it indeed relevant? The evaluation algorithm does not seem to depend on model structure.":** The reviewer is correct; in this context, "large-scale" refers to the fact that generalization gaps in our non-asymptotic bounds diminish as network size increases. For a given maximum tolerable generalization gap, the minimum network size can be explicitly determined.
>
> - **"Text in Figure 3 is too small to read.":** We apologize for this issue. In the revised version, this figure will be presented in full-scale in the Appendix section.
>
> We hope that these responses address the reviewer's concerns and respectfully request that they consider revisiting their score. We believe the current score does not fully reflect the contributions and effort in our work.

---

> > ### Author Response · Authors · 2024-11-22
> >
> > Dear Reviewer Vduy,
> >
> > Since we are approaching the end of the discussion period, we appreciate it if you could give us your feedback after reading the rebuttal.
> >
> > Thanks in advance,
> > Authors

---

> > ### Comment · Reviewer_Vduy · 2024-11-25
> >
> > Thank you for your rebuttal. I have increased my score, but I still lean towards rejection. The authors emphasize the difference on client distribution and meta distribution, but I am not totally convinced. It feels like with high probability client data distributions are close. Also I do not see the advantage of posing assumptions on the meta-distribution over on client distributions.

---

> > > ### Author Response · Authors · 2024-11-27
> > >
> > > Let us explore this issue further with an example. To make the concept clearer, we begin with an example at a lower modeling level.
> > >
> > > Assume $X_1, \ldots, X_n$ are sampled from "any" distribution, for example a Gaussian distribution with unknown mean and variance (we make no assumption regarding the type of distributions). As a result, the $X_i$ values are not necessarily close to each other. Now, assume $Y_1, \ldots, Y_m$ are sampled (independently of the $X_i$ values) from another unknown distribution. The only information we have is that the second distribution is similar to the first one, for example, another Gaussian distribution with a mean and/or variance that is close to those of the first Gaussian. Again, the $Y_i$ values are not necessarily close to each other, nor are they necessarily close to the $X_i$ values. The similarity pertains to the distributions themselves rather than the individual samples.
> > >
> > > Our work operates at a higher level. We assume that distributions $P_1, \ldots, P_n$ are sampled from a meta-distribution $\mu$, and that other distributions $Q_1, \ldots, Q_m$ are sampled from a different meta-distribution $\mu'$. The similarity here is at the level of the meta-distributions. However, neither the $P_i$ distributions nor the $Q_i$ distributions are necessarily similar to one another.

---

### Official Review · Reviewer_w4NP · 2024-11-04

**Soundness:** 3
**Presentation:** 2
**Contribution:** 3
**Rating:** 5
**Confidence:** 3

**Summary:**

This paper studies the generalization capabilities of federated models against unseen clients. Specifically, they aim to provide certified guarantees for the model's performance on a different and unseen target network. Theoretical analysis reveals the importance of the obtained results, where the deviation between different networks is bounded by $f$-divergence or Wasserstein distance.

**Strengths:**

* The topic is important for the real-world deployment of federated learning models.
* The idea is intuitive and enjoys strong guarantees.
* Extensive experiments have been provided.

**Weaknesses:**

* The paper studies federated learning under IID distribution, while the current real-world application leans more toward non-IID clients. In the Introduction, the authors have used many examples to motivate the generalization capabilities of federated learning, which fundamentally stems from non-IID data.
* The presentation of the paper can be improved. For example, the contributions can be listed as bullet points in the Introduction.
* The third part of the related work is confusing. The study of the paper is on IID data, as illustrated in Section 4. Yet, the authors discuss intensively federated learning algorithms under non-IID data. Moreover, the discussion mingles personalized federated learning with non-personalized ones without distinctions.
* Theorem 5.2 is a strong guarantee in the sense that the generalization gap does not produce a non-vanishing term related to $\epsilon$. However, the authors fail to reason the root cause behind it.
* The authors claim that $\hat{B}^{*}(\epsilon)$ is asymptotically minimax optimal. Yet, Theorem 5.2 is a one-sided bound. To claim it is asymptotically optimal, both lower and upper bounds have to be provided. The same concern goes to Theorem 6.2.
* The authors present extensive experiment results, yet, they do not provide codes to reproduce them.

**Questions:**

* Is it possible to extend the results to non-IID setups? What would be the potential obstacles?
* Can the authors discuss why no non-vanishing term related to $\epsilon$ appears in Theorem 5.2? In addition, what conventional bounds are the authors referring to?
* Can the authors elaborate more on the minimax optimal bounds?

---

> ### Author Response · Authors · 2024-11-16
> **Response to Reviewer w4NP**
>
> We thank the reviewer for their time and thoughtful feedback. Below, we address each of the reviewer's concerns in detail.
>
> 1. **"The paper studies federated learning under IID distribution":** We believe there may be a misunderstanding regarding our data generation setting. Our work explicitly considers a non-IID scenario, where different clients have distinct data distributions. We make no assumptions about the nature or similarity of these distributions. In the federated learning (FL) literature, "non-IID" refers to cases where client distributions differ (as in our setting), not to cases where samples within each client are dependent or from varied distributions.
>
>     Our only assumption is that samples within each client are drawn i.i.d. from its specific distribution (a standard assumption in non-IID FL), and that clients are sampled from a broader meta-distribution. Notably, without this meta-distribution assumption, providing any theoretical guarantees for the target network becomes mathematically infeasible.
>
> 2. **"The presentation of the paper can be improved":** We will improve the presentation to enhance clarity.
>
> 3. **"The third part of the related work is confusing. The study of the paper is on IID data...":** As noted, our study addresses a non-IID setting in which clients possess distinct (and potentially distant) distributions. This issue has been clarified above in response to the first comment.
>
> 4. **"Theorem 5.2 is a strong guarantee in the sense that the generalization gap does not produce a non-vanishing term related to $\epsilon$. However, the authors fail to reason the root cause behind it.":** This result is consistent with robust optimization theory when carefully developed. The generalization gap vanishes as sample or network size grows because the empirical part is adversarially computed, inflating its value. Increasing $\epsilon$ affects the empirical part, not the generalization gap, and this characteristic is highly advantageous for models/distribution pairs with inherent robustness.
>
> 5. **"The authors claim that $\hat{B}^{*}(\epsilon)$ is asymptotically minimax optimal. But there are no proofs":** Our claim is that the bounds are "asymptotically" minimax optimal, meaning they become optimal as $K$ and $\min_k n_k$ approach infinity. The proof, while straightforward, was omitted for brevity: By the strong law of large numbers, our empirical bound converges (almost surely) to the worst-case adversarial loss/CDF computed based on a meta-distribution deviating at most $\epsilon$ from $\mu$. We will add supporting theorems to clarify this claim.
>
> 6. **"The authors present extensive experimental results, yet they do not provide codes to reproduce them.":** The reviewer is correct, and we apologize for the oversight. We will include code for reproducibility in the supplemental material. However, please note that our paper is predominantly theoretical, with experiments intended to demonstrate the practical computability of our bounds.
>
> **Questions:** We believe these questions are answered within the responses above.
>
> We hope the reviewer finds our responses satisfactory and would appreciate it if they could reconsider their score.

---

> > ### Author Response · Authors · 2024-11-22
> >
> > Dear Reviewer w4NP,
> >
> > Since we are approaching the end of the discussion period, we appreciate it if you could give us your feedback after reading the rebuttal.
> >
> > Thanks in advance,
> > Authors

---

> > > ### Comment · Reviewer_w4NP · 2024-11-24
> > >
> > > Thank you for the responses. After reading the authors' replies and the reviewer cohorts' comments, I have decided to keep my score.

---

### Author Response · Authors · 2024-11-16
**Global Response and Some Major Clarifications**

Dear reviewers,
Thank you for taking the time to review our work.

We believe there may have been some misunderstandings regarding our paper, particularly in the reviews provided by Reviewers w4NP and Vduy. To clarify, we would like to restate the key aspects of our data generation process:

- We have considered a **non-IID** setting where each client in the network possesses a unique and distinct data distribution.
These distributions can vary significantly between clients, and we have not made any assumptions about their similarity or closeness.

- **Within** each client, data samples are i.i.d. with respect to their corresponding data distribution. This is a standard assumption in almost all non-IID federated learning methods.

- Our "only" assumption is that individual client distributions can be treated as independent samples drawn from a broader meta-distribution (a distribution over distributions). Importantly, this does not imply that all clients share the same internal data distribution or that their distributions are close, in any shape or form. Furthermore, we have made no specific assumptions about the characteristics of this meta-distribution.

- $\varepsilon$-proximity in our work applies at the level of meta-distributions which correspond to two different networks, and not the individual data distributions which underlie user samples. For example, consider the city of New York as a meta-distribution and its diverse and various citizens as data distributions which generate data samples (e.g. images) in their cell-phones by taking pictures. Hong Kong can be considered as a different meta-distribution, where its residents can have slightly different habits in taking images or selfies. Therefore, the two meta-distributions are shifted w.r.t. each other. We want to provide performance guarantees on the average and CDF of users in Hong Kong, via only having access to users from New York.

We have also provided detailed, point-by-point responses to each of your comments and concerns.

Thank you in advance for your consideration.

---

### Meta-Review · Area_Chair_x3cN · 2024-12-20

**Metareview:**

a) Summary

This paper develops theoretical guarantees for evaluating machine learning models on unseen federated networks where client data distributions are heterogeneous and sampled from distinct meta-distributions. It derives robust bounds on the empirical average loss and loss CDF using Wasserstein distance and f-divergence, demonstrating that generalization gaps vanish as the network size and client sample sizes increase. The findings are validated empirically, showcasing the bounds' tightness and practicality, though real-world applicability and comparisons to existing methods are underexplored.

b) Strengths

- Theoretical Contributions: the idea of protecting privacy by adversarially perturbing losses is novel and interesting. So are the adversarially robust extensions of the Glivenko-Cantelli theorem and DKW inequality under meta-distribution shifts. These bounds are proven to be tight, minimax optimal, and non-asymptotic, ensuring vanishing generalization gaps with increasing clients and sample sizes.

- Extensive experimental validation: Extensive experiments are included, though they are primarily illustrative of the theoretical results

c) Weakenesses

- Practical Relevance: The biggest weakness is about the real-world applicability of the theoretical results, particularly under non-IID data setups, as were raised (Vduy, w4NP). Assumptions such as bounded density ratios and the use of f-divergence are perceived as either unclear or overly restrictive.

- Presentation: Some reviewers found the paper challenging to follow, with critical definitions and assumptions deferred to the appendix (oDR9) and difference between meta-distributions and individual client distributions confusing.

- Lack of comparisons and baselines: The paper does not compare its bounds directly with existing works.

d) Reason for **rejection**

The paper presents strong theoretical contributions, including novel bounds for federated model evaluation under meta-distributional shifts, but its real-world applicability is unclear due to restrictive assumptions. Key issues include poor presentation of definitions and assumptions, lack of experimental comparisons to existing methods, and mixed reviewer feedback indicating concerns about clarity and maturity. While promising, the work is not yet ready for publication.

**Additional Comments On Reviewer Discussion:**

During the rebuttal period, the reviewers raised concerns about unclear assumptions (e.g., meta-distribution similarity and bounded density ratios), presentation issues, and limited practical relevance of the theoretical results.

The authors clarified key misunderstandings, such as distinguishing meta-distributional shifts from individual client distribution shifts, and explained the role of assumptions like bounded density ratios. They also promised clearer definitions, expanded literature comparisons, and better experimental details in a revised version.

While some points were satisfactorily addressed (e.g., theoretical clarifications), concerns about real-world applicability, presentation quality, and experimental validation persisted. These unresolved issues were weighted heavily in the final decision to recommend rejection.

---

### Decision · Program_Chairs · 2025-01-22

Reject